# Targeted apoptosis of macrophages and osteoclasts in arthritic joints is effective against advanced inflammatory arthritis

Caifeng Deng [1,2], Quan Zhang[3,4], Penghui He[1], Bin Zhou[2,5], Ke He[2,5], Xun Sun [1], Guanghua Lei [2,5,6✉], Tao Gong [1✉] & Zhirong Zhang[1]

Insufficient apoptosis of inflammatory macrophages and osteoclasts (OCs) in rheumatoid arthritis (RA) joints contributes toward the persistent progression of joint inflammation and destruction. Here, we deliver celastrol (CEL) to selectively induce apoptosis of OCs and macrophages in arthritic joints, with enzyme-responsive nanoparticles (termed PRNPs) composed of RGD modified nanoparticles (termed RNPs) covered with cleavable PEG chains. CEL-loaded PRNPs (CEL-PRNPs) dually target OCs and inflammatory macrophages derived from patients with RA via an RGD-αvβ3 integrin interaction after PEG cleavage by matrix metalloprotease 9, leading to increased apoptosis of these cells. In an adjuvant-induced arthritis rat model, PRNPs have an arthritic joint-specific distribution and CEL-PRNPs efficiently reduce the number of OCs and inflammatory macrophages within these joints. Additionally, rats with advanced arthritis go into inflammatory remission with bone erosion repair and negligible side effects after CEL-PRNPs treatment. These findings indicate potential for targeting chemotherapy-induced apoptosis in the treatment of advanced inflammatory arthritis.

[1] Key Laboratory of Drug-Targeting and Drug Delivery System of the Education Ministry, Sichuan Engineering Laboratory for Plant-Sourced Drug and Sichuan Research Center for Drug Precision Industrial Technology, West China School of Pharmacy, Sichuan University, Chengdu 610064, China. [2] Department of Orthopaedics, Xiangya Hospital, Central South University, Changsha 410008, China. [3] Institute of Materia Medica, School of Pharmacy, Chengdu Medical College, Chengdu 610500, China. [4] Development and Regeneration Key Lab of Sichuan Province, Department of Pathology, Department of Anatomy and Histology and Embryology, Chengdu Medical College, Chengdu 610500, China. [5] Hunan Key Laboratory of Joint Degeneration and Injury, Changsha 410008, China. [6] National Clinical Research Center of Geriatric Disorders, Xiangya Hospital, Central South University, Changsha 410008, China. ✉email: lei_guanghua@csu.edu.cn; gongtaoy@126.com

Rheumatoid arthritis (RA) is a chronic autoimmune inflammatory disease characterized by synovial inflammation and joint destruction[1]. Synovial inflammation is the dominant feature in the early stage of RA[2–4]. In addition to the inflammatory symptoms of joint swelling and synovial inflammation, obvious cartilage damages and bone erosion occur during the persistent progression of RA, which can eventually result in joint deformity and disability[5–7]. Patients with advanced RA suffer from the loss of physical function and low quality of life[7,8]. Therefore, a rational therapeutic approach that can alleviate synovial inflammation and reverse bone erosion is urgently needed for the treatment of advanced RA.

Macrophages and osteoclasts (OCs) have been demonstrated to play key roles in the pathogenesis of RA. The increase in the abundance of synovial macrophages is an early hallmark of rheumatic disease. Synovial macrophages from RA patients show distinct activation states and represent one potential key mediator of joint inflammation[3,9,10]. Additionally, studies have revealed large numbers of OCs at sites of arthritic bone erosion[11–13]. OCs are terminally differentiated cells with the unique ability to resorb bone matrix[14–16]. Notably, OCs in RA-affected joints can accelerate synovial inflammation through the production of pro-inflammatory cytokines[17,18]. Both OCs and synovial inflammatory macrophages express high levels of $\alpha v\beta 3$ integrin. $\alpha v\beta 3$ integrin has been demonstrated to play important role in activated macrophage-dependent inflammation and OC-dependent bone resorption[19,20]. Generally, the life spans of macrophages and OCs are precisely regulated by apoptosis to maintain immune homeostasis and bone function balance, respectively[21–24]. However, macrophages and OCs from RA joints show decreased apoptotic rates compared with those from healthy controls[21,25,26]. The insufficient apoptosis of macrophages and OCs in the RA joint contributes toward the persistent progression of joint inflammation and joint destruction. Accordingly, inducing the apoptosis of both macrophages and OCs in RA joints is a promising strategy for advanced RA therapy.

Current antirheumatic drugs, including glucocorticoids and biological antibodies, mainly target the macrophages-induced inflammatory response to reduce synovial inflammation[5,27]. However, the application of glucocorticoids can result in severe side effects including bone loss and hyperglycemia[28,29], while the frequent use of antibodies to block inflammatory cytokines can cause systemic immune suppression, thereby leading to a high risk of infections[30,31]. Furthermore, not all patients respond to antibody therapy and the benefits of using antibody therapy is short-lived[21,30,32]. Recently, Janus Kinase (JAK) inhibitors have been reported to have the potential to reverse the bone erosions in RA[33,34], yet JAK inhibition could lead to serious and opportunistic infections[33,35]. Celastrol (CEL), a cytotoxic chemotherapeutic drug, can induce apoptosis in tumor cells and has been widely studied in cancer therapy[36,37]. In addition to its known efficacy in cancer treatment, CEL was previously shown in our laboratory to also treat glomerulonephritis by inducing the apoptosis of mesangial cells[38]. In a rat model of anti-Thy1.1 nephritis, the targeted delivery of CEL to the mesangial cells significantly increased its therapeutic efficacy and decreased its side effects[38]. Inspired by our previous findings, we hypothesized that selectively delivering CEL to both macrophages and OCs in RA joints may efficiently induce apoptosis in these cells, thus reducing synovial inflammation and reversing bone erosion in advanced RA.

In this study, we report the development of matrix metalloproteinase 9 (MMP9)-cleavable, polyethylene glycol (PEG)- and RGD peptide-modified poly (D, L-lactide-co-glycolide) (PLGA) nanoparticles (termed "PRNPs") for the targeted delivery of CEL to both OCs and macrophages in arthritic joints. PRNPs show

high cellular uptake in both OCs and inflammatory macrophages derived from patients with late-stage RA via RGD-$\alpha v\beta 3$ integrin interaction after responding to MMP9. In addition, CEL-loaded PRNPs (CEL-PRNPs) in the presence of MMP9 effectively induce the apoptosis of these cells. In adjuvant-induced arthritis (AIA) rats, PRNPs show an arthritic joints-specific distribution and target both OCs and macrophages within these joints. Further, the intravenous administration of CEL-PRNPs effectively relieves the ankle and paw swelling, restores the balance of bone function, and reverses bone erosion in the inflamed joints of AIA rats with advanced arthritis. Of note, CEL-PRNPs seem safe and induce negligible apoptosis in normal organs. In summary, CEL-PRNPs show great promise in promoting inflammatory remission and bone erosion repair in advanced inflammatory arthritis.

## Results

**Features of pathology in different stages of arthritis**. AIA in the rat is a well-characterized model for assessing the stages of pathology in RA and exploring arthritis mechanisms[39–41]. Using micro-computed tomography (micro-CT) and histological assays, we investigated joint destruction and synovial inflammation in AIA rats. Micro-CT was used to scan the ankle joints of both normal rats and AIA rats. We found that, in contrast with the smooth bone surface and negligible bone erosion in ankle joints from rats with early-stage arthritis, AIA rats with advanced arthritis displayed rough bone surfaces, severe bone erosion, and significantly decreased bone mineral density (BMD) (Fig. 1a–c). Applying tartrate-resistant acid phosphatase (TRAP) staining of OCs further demonstrated that there were large numbers of OCs and areas of bone erosion in AIA rats within the advanced arthritis group (Fig. 1a). We also examined cartilage integrity using safranin O and toluidine blue staining to label the glycosaminoglycans (GAGs) in cartilage tissues[42,43]. The cartilage of AIA rats with early-stage arthritis remained intact and was comparable to that of the normal group. However, significant reductions of safranin O- and toluidine blue-positive areas were observed on the cartilage surface of AIA rats with advanced arthritis, indicating that a loss of GAGs and cartilage destruction occurred in advanced arthritis (Supplementary Fig. 1). We next investigated levels of synovial inflammation in AIA rats using immunohistochemical staining. Synovial macrophages with high potential to induce inflammation were identified as CD68+ cells[44–46]. Immunohistochemical images revealed that rats with disease induction showed increased infiltration of CD68+ macrophages into the synovium compared with the normal group (Fig. 1a). Notably, AIA rats with advanced arthritis showed an excessive abundance of CD68+ macrophages and synovial hyperplasia, consistent with the results of hematoxylin and eosin (H&E) assay (Fig. 1a; Supplementary Fig. 1). In addition to the increased accumulation of CD68+ macrophages, the augmented expression of MMP9 in the arthritic inflammatory environment was also observed during the induction of AIA (Fig. 1a). Previous studies also demonstrated macrophages and OCs promoted the high expression of MMP9 in the inflammatory microenvironment of RA[47,48]. Thus, these results revealed that there were excessive levels of CD68+ macrophages and OCs in the arthritic joints of AIA rats with advanced arthritis.

**Preparation and characterization of CEL-PRNPs**. PLGA nanoparticles exhibit good biocompatibility and high drug loading efficacy and have been widely employed as a drug delivery system[49–51]. Therefore, PLGA nanoparticles were adopted as the drug carrier in this work. We first modified PLGA with RGD peptide *via* the maleimide-thiol coupling reaction between Cys-RGD and Mal-PEG-PLGA and the chemical structure was

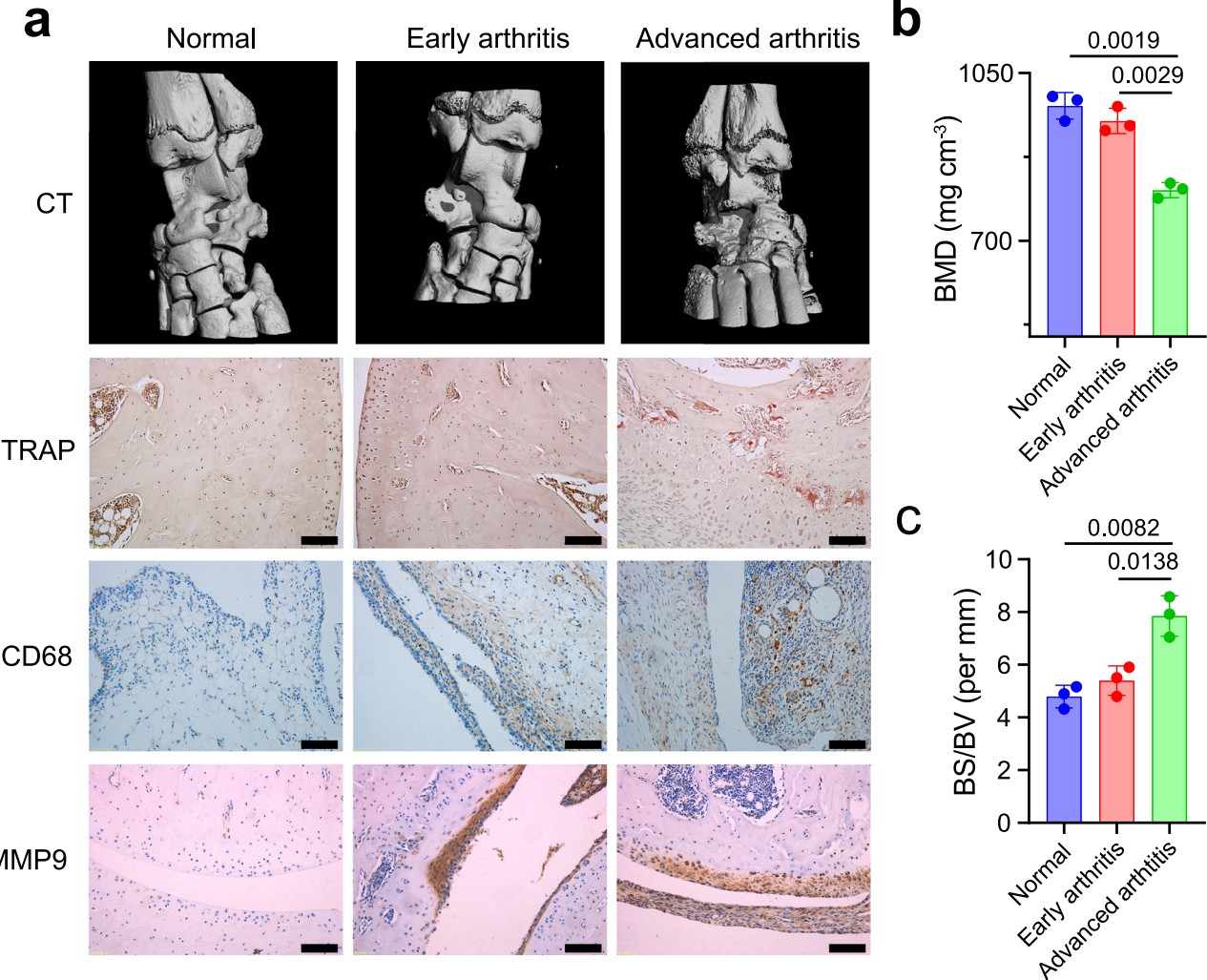

**Fig. 1 Pathological features differ between rats with early and advanced arthritis. a** OCs-induced bone erosion and macrophages-mediated synovial inflammation in advanced arthritis. Representative micro-CT images of ankle joints showing bone erosion levels in ankle joints from normal rats, AIA rats with early-stage arthritis, and AIA rats with late-stage arthritis. Immunohistochemical analyses of the TRAP (tartrate-resistant acid phosphatase)-stained OCs, CD68+ synovial macrophages, and MMP9 (matrix metalloproteinase 9) expression in the joint tissues from rats in each group ($n = 3$). Scale bar = 100 μm. **b**, **c** Quantitative micro-CT analyses of bone mineral density (BMD) and bone surface density (BS/BV). Data represent the mean ± SD ($n = 3$ independent animals). Statistical significance was determined by a two-sided Student's $t$ test.

analyzed by proton nuclear magnetic resonance ($^1$H-NMR) spectroscopy (Supplementary Fig. 2). Different PLGA nano-particles were prepared as shown in Fig. 2a. Briefly, CEL-loaded PLGA nanoparticles (CEL-NPs) and CEL-loaded, RGD-modified PLGA nanoparticles (CEL-RNPs) were both prepared using an emulsion/solvent evaporation method. To obtain CEL-PRNPs, PEG$_{2000}$-MMP9 cleavable peptide was linked to CEL-RNPs using the water phase reaction method[52]. The particle sizes of CEL-NPs and CEL-RNPs were 155.7 ± 4.9 nm and 154.1 ± 4.6 nm, respectively. After covered with cleavable PEG$_{2000}$, the particle size of CEL-PRNPs slightly increased to 162.2 ± 6.6 nm (Fig. 2b; Supplementary Table 1). The average zeta potentials of CEL-RNPs and CEL-PRNPs were −3.2 ± 0.6 mV and −5.3 ± 0.4 mV, respectively. The CEL encapsulation efficiencies of various developed PLGA nanoparticles were all close to 90% (Supplementary Table 1). The transmission electron microscope (TEM) images demonstrated that the morphologies of the prepared CEL-PRNPs were generally spherical and uniformly dispersed (Fig. 2c). The serum stability assay was conducted to investigate interactions between various developed PLGA nanocarriers and blood components. The average particle sizes remained nearly

unchanged for CEL-NPs, CEL-RNPs, and CEL-PRNPs after 24 h of storage in 10% fetal bovine serum (FBS) at 37 °C (Fig. 2d; Supplementary Fig. 3), suggesting the good stability of these nanoparticles in serum.

**CEL-PRNPs increase apoptosis of OCs and inflammatory macrophages.** To investigate whether PRNPs could target OCs and pathogenic macrophages via RGD-mediated endocytosis, a cellular uptake study was performed. OCs were established by using M-CSF- and RANKL-stimulated bone marrow macrophages (BMMs) and pathogenic macrophages were obtained by using lipopolysaccharide (LPS)-activated BMMs[53]. The successful genesis of OCs was confirmed using TRAP staining. Results showed that RANKL-induced OCs stained red and were multi-nuclear, while the unstimulated BMMs stained yellow (Supplementary Fig. 4), suggesting that OCs were efficiently generated via RANKL stimulation. To investigate the distribution behaviors of various prepared nanoparticles in OCs and LPS-activated macrophages, the fluorescent probe, coumarin 6 (C6), was loaded into the nanoparticles. Confocal images of the OCs and LPS-activated macrophages showed that RNPs conferred significantly

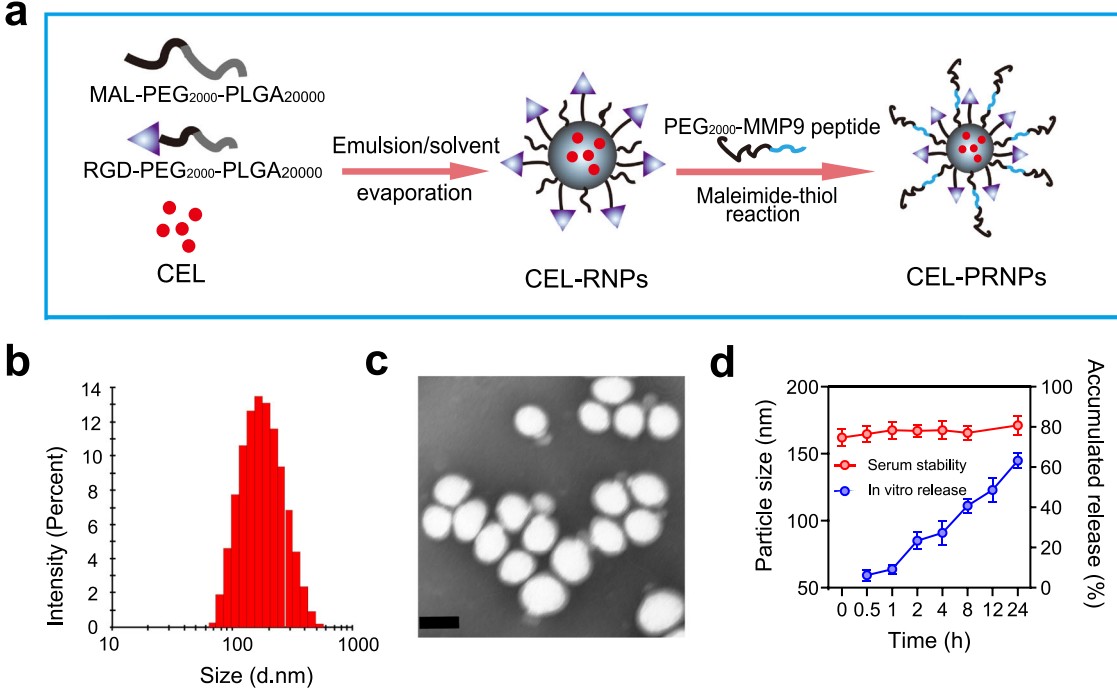

**Fig. 2 Preparation and characterization of CEL-PRNPs. a** Schematic illustration of CEL-PRNPs preparation. CEL celastrol, MMP9 matrix metalloproteinase 9, CEL-RNPs CEL-loaded RGD peptide-modified poly (D, L-lactide-co-glycolide) (PLGA) nanoparticles, CEL-PRNPs CEL-loaded MMP9-cleavable polyethylene glycol (PEG)- and RGD peptide-modified PLGA nanoparticles. **b** Representative size distribution image of CEL-PRNPs ($n = 3$ independent samples). **c** TEM image of CEL-PRNPs ($n = 3$ independent samples). Scale bar = 100 nm. **d** The serum stability of CEL-PRNPs during 24 h incubation with 10% FBS at 37 °C and cumulative CEL release from CEL-PRNPs in PBS at 37 °C. Data represent mean ± SD ($n = 3$ independent samples).

increased green fluorescence signal to these cells compared with NPs (Fig. 3a; Supplementary Figs. 5 and 6). However, this effect was not observed in the non-activated BMMs (Supplementary Fig. 7). These results suggested that the RNPs could selectively target both OCs and LPS-activated macrophages for cellular uptake. In the absence of MMP9, PRNPs showed remarkably reduced cellular uptake, whereas, in the presence of MMP9, this effect was reversed (Fig. 3a; Supplementary Figs. 5 and 6). This indicated that the PEG chains on the PRNPs could be cleaved by MMP9, thereby exposing the RGD peptide for the selective tar-geting of the OCs and LPS-activated macrophages. The results of quantitative cellular uptake analysis based on flow cytometry showed the same trends (Fig. 3d, e). Furthermore, the dual-targeting ability of PRNPs was also investigated on human OCs and inflammatory macrophages. Human OCs were obtained by using M-CSF- and RANKL-stimulated peripheral blood mono-nuclear cells from patients with late-stage RA. Human inflam-matory macrophages were isolated by magnetic-activated cell sorting method from synovial tissues of patients with late-stage RA undergoing joint replacement surgery. As shown in Fig. 4a–c, the increased distribution of RNPs and PRNPs (in the presence of MMP9) were also observed on OCs and inflammatory macro-phages derived from patients with late-stage RA. These results demonstrated that RNPs and PRNPs (in the presence of MMP9) could target both OCs and inflammatory macrophages through ligand–receptor interactions.

The inadequate apoptosis of macrophages and OCs in the rheumatoid inflammatory microenvironment is an important pathomechanism in synovial hyperplasia and joint destruction[21,54,55]. To investigate whether CEL-PRNPs could effectively induce the apoptosis of OCs and inflammatory macrophages, their apoptotic profile was determined by flow cytometry assay. As shown in Fig. 3b, f, g, CEL-RNPs treatment significantly increased the apoptotic cell percentages in both OCs

and LPS-activated macrophages compared with the CEL-NPs and CEL-PRNPs. However, in the presence of MMP9, treatment with the CEL-PRNPs resulted in high levels of apoptosis in both OCs and LPS-activated macrophages, demonstrating that CEL-PRNPs were MMP9-responsive. As mitochondrial dysfunction is also a hallmark of apoptosis, we further measured the mitochondrial membrane potential (a marker of mitochondrial dysfunction) in macrophages with the JC-1 dye[56]. The JC-1 dye tends to aggregate (with red fluorescence) in normal mitochondria, and its color changes from red to green when the membrane potential collapses[57]. Confocal images showed that both CEL-RNPs and CEL-PRNPs (in the presence of MMP9) resulted in significantly higher green fluorescence intensity signal compared with CEL-NPs (Fig. 3c), suggesting the severe disruption of the mitochon-drial membrane in LPS-activated macrophages. Furthermore, CEL-RNPs and CEL-PRNPs (in the presence of MMP9) also triggered the higher rates of apoptosis among OCs and inflammatory macrophages derived from patients with late-stage RA, when compared with CEL-NPs and CEL-PRNPs (in the absence of MMP9) (Fig. 4e, f). These results proved that CEL-PRNPs had excellent ability to cause the apoptosis of OCs and inflammatory macrophages after responding to MMP9.

**PRNPs target both OCs and inflammatory macrophages in arthritic joints**. We studied the biodistribution of the PRNPs in AIA rats with advanced arthritis using an in vivo imaging system and PRNPs loaded with DiD, a fluorescent probe. The in vivo fluorescence images showed that PRNPs significantly increased the DiD fluorescence distribution in inflamed joints compared with NPs and RNPs (Fig. 5a, d; Supplementary Fig. 8). Of note, the use of PRNPs remarkably reduced the DiD fluorescence distribution in the lung, liver and spleen compared with RNPs (Fig. 5a, e), which indicated that the nonspecific distribution of RNPs was avoided through the use of a surface coating

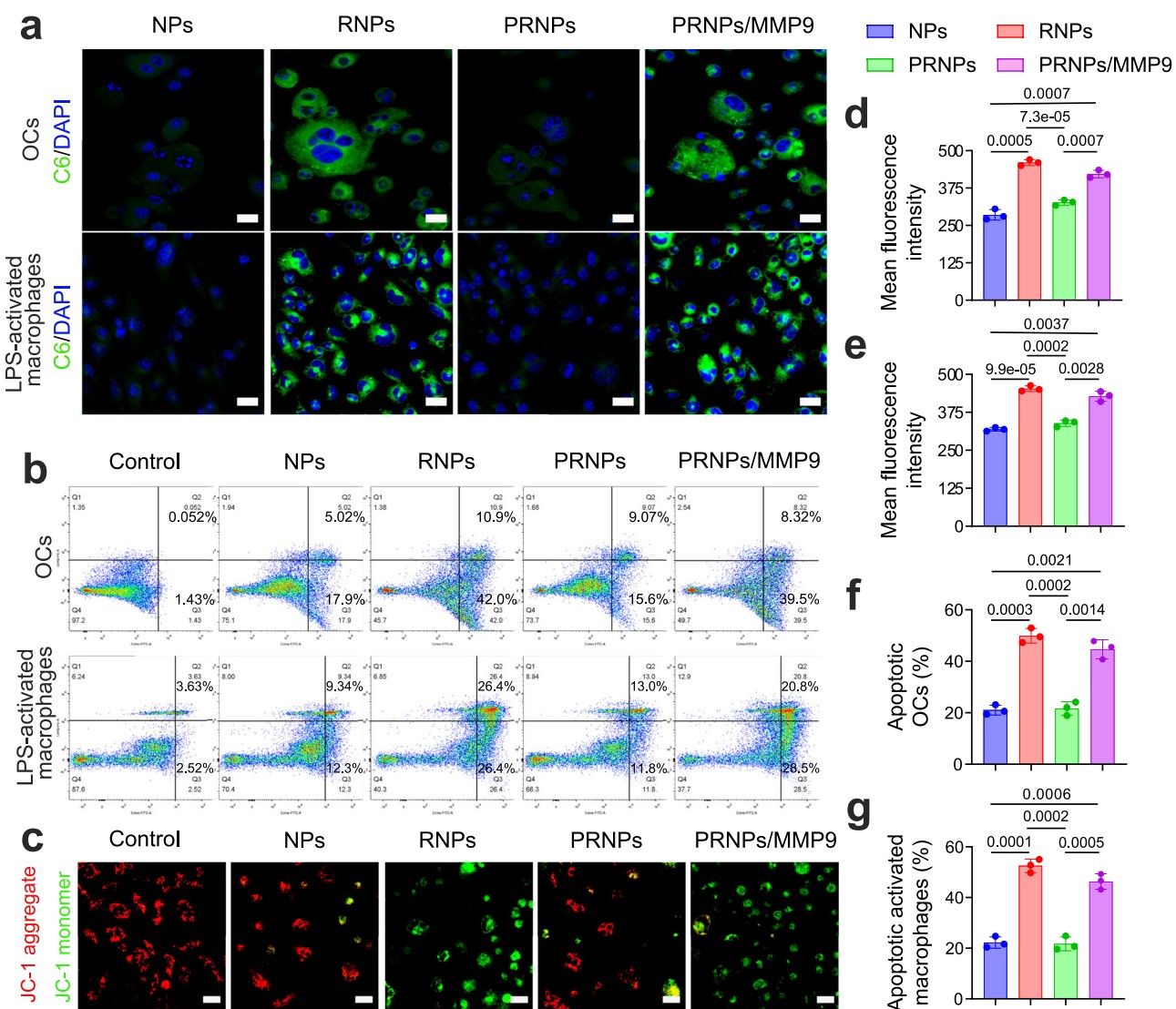

**Fig. 3 Increased apoptosis of OCs and LPS-activated macrophages caused by RGD-mediated endocytosis of PRNPs. a** Confocal images of cellular uptake in OCs and LPS-activated macrophages ($n = 3$ independent samples). Scale bar = 50 μm. **b** Flow cytometric analysis of OCs and LPS-activated macrophages apoptosis induced by CEL-RNPs, CEL-RNPs, or CEL-PRNPs with or without the presence of MMP9 for 24 h at the CEL concentration of 100 ng/mL. **c** Confocal images showing JC-1 assay of LPS-activated macrophages treated with CEL-RNPs, CEL-RNPs, or CEL-PRNPs with or without the presence of MMP9 for 24 h at the CEL concentration of 100 ng/mL ($n = 3$ independent samples). Scale bar = 50 μm. **d, e** Quantitative cellular uptake C6-loaded NPs, C6-loaded RNPs, or C6-loaded PRNPs on OCs (**d**) and LPS-activated macrophages (**e**) after 1 h incubation at the C6 concentration of 50 ng/mL. Data represent mean ± SD ($n = 3$ independent samples). Statistical significance was determined by a two-sided Student's $t$ test. **f, g** Quantitative analysis for the apoptosis of OCs (**f**) and LPS-activated macrophages (**g**) by CEL-NPs, CEL-RNPs, or CEL-PRNPs. Data represent mean ± SD ($n = 3$ independent samples). Statistical significance was determined by a two-sided Student's $t$ test. OCs osteoclasts, LPS lipopolysaccharide, C6 coumarin 6, DAPI 2-(4-Amidinophenyl)-6-indolecarbamidine dihydrochloride, MMP9 matrix metalloproteinase 9, NPs poly (D, L-lactide-co-glycolide) (PLGA) nanoparticles, RNPs RGD peptide-modified PLGA nanoparticles, PRNPs MMP9-cleavable polyethylene glycol (PEG)- and RGD peptide-modified PLGA nanoparticles.

comprising cleavable PEG chains, as ανβ3 integrins are also highly expressed in normal tissues[58,59]. The ex vivo fluorescence images also demonstrated that PRNPs had the longest blood circulation time (Fig. 5a, e), which further suggested that the PEG$_{2000}$-MMP9 cleavable peptide had been successfully linked to RNPs. Interestingly, PRNPs accumulation in inflamed joints at 24 h after intravenous injection was still as high as that in inflamed joints at 2 h (Supplementary Fig. 8). However, their accumulation in normal organs at 24 h was obviously less than that at 2 h (Supplementary Fig. 9). These results illustrated that PRNPs displayed the inflamed joint-targeting ability and extended retention in inflamed joints.

Next, AIA rats with unilateral inflamed joints were adopted to evaluate the selective accumulation of PRNPs in inflamed joints compared with normal joints. As shown in Fig. 5b, c, f, there were no significant differences in the distribution of free DiD between inflamed joints and non-inflamed joints. In contrast, DiD-loaded NPs, RNPs, and PRNPs showed a higher accumulation in inflamed joints than non-inflamed joints. As expected, PRNPs displayed the highest distribution in inflamed joints among the three nanoparticle types, which was consistent with the results obtained for AIA rats with bilateral inflamed joints (Fig. 5a, d).

The above results demonstrated that PRNPs increased drug distribution in arthritic joints. To investigate whether PRNPs could

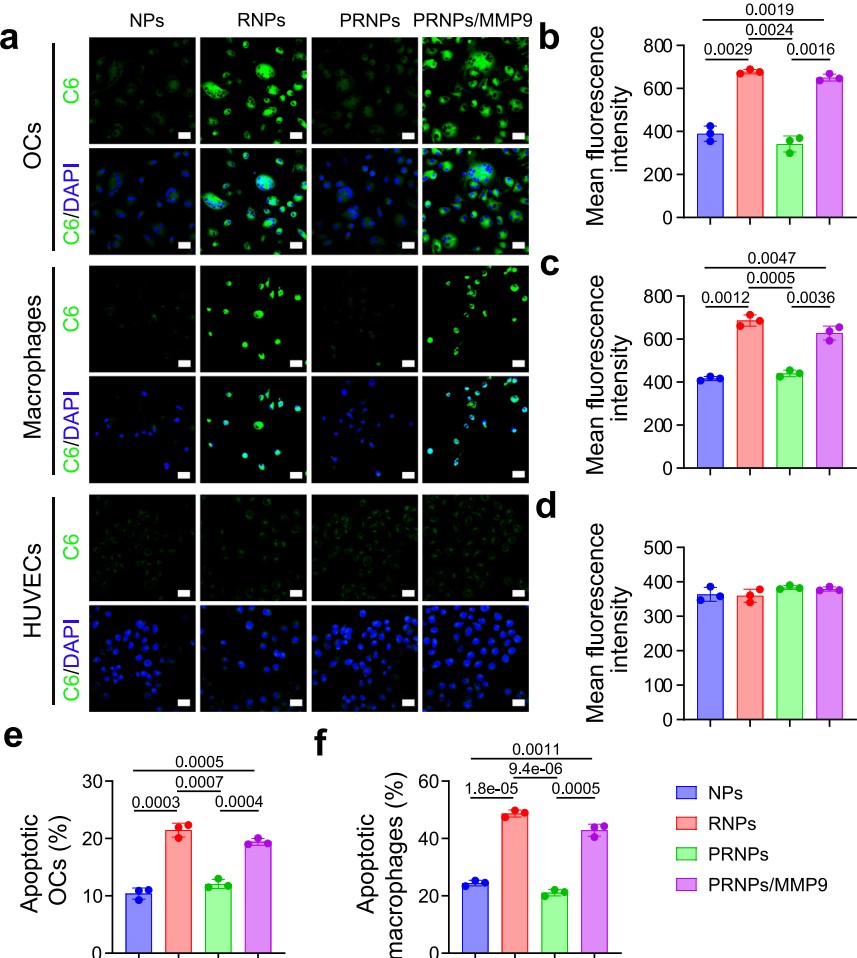

**Fig. 4 Increased apoptosis of OCs and synovial macrophages derived from patients with late-stage RA by PRNPs. a** Confocal images of the cellular uptake on OCs and synovial macrophages ($n = 3$ independent samples). Scale bar = 50 μm. **b–d** Quantitative analysis of the cellular uptake of C6-loaded NPs, RNPs or PRNPs on OCs (**b**), synovial macrophages (**c**), and HUVECs (**d**) after 1-h incubation at the C6 concentration of 50 ng/mL. Data represent mean ± SD ($n = 3$ independent samples). Statistical significance was determined by a two-sided Student's *t* test. **e, f** Quantitative analysis for the apoptosis of OCs (**e**) and synovial macrophages (**f**) by CEL-NPs, CEL-RNPs, or CEL-PRNPs. Data represent mean ± SD ($n = 3$ independent samples). Statistical significance was determined by a two-sided Student's *t* test. OCs osteoclasts, C6 coumarin 6, DAPI 2-(4-Amidinophenyl)-6-indolecarbamidine dihydrochloride, MMP9 matrix metalloproteinase 9, HUVECs human umbilical vein endothelial cells, NPs poly (D, L-lactide-co-glycolide) (PLGA) nanoparticles, RNPs RGD peptide-modified PLGA nanoparticles, PRNPs MMP9-cleavable polyethylene glycol (PEG)- and RGD peptide-modified PLGA nanoparticles.

target both OCs and inflammatory macrophages in arthritic joints, the distribution of different DiD formulations in both types of cells was determined using the immunofluorescent staining method. Inflammatory macrophages and OCs were determined by immunofluorescence analysis of CD68 and CD51 (green fluorescence). As shown in Fig. 6, the DiD fluorescence distribution of PRNPs in the synovial joint was the highest among the three nanoparticle types, which was consistent with the results of in vivo and ex vivo imaging studies. In addition, DiD solution and DiD labeled NPs showed low levels of colocalization of the red (DiD) and green fluorescence, suggesting the nonspecific distributions in inflamed joints. Whereas, the DiD fluorescence of RNPs and PRNPs was mainly overlapped with the green fluorescence in synovial tissues. Therefore, the results proved that PRNPs could detach their PEG chains and transform into RNPs within inflamed joints, so as to efficiently and selectively deliver drugs into OCs and macrophages located in inflamed joints.

**CEL-PRNPs decrease the number of OCs and macrophages in arthritic joints**. The above results showed that PRNPs exhibited good selectivity toward both OCs and inflammatory macrophages

in arthritic joints. Furthermore, CEL-PRNPs (in the presence of MMP9) could more effectively induce apoptosis in both OCs and inflammatory macrophages due to the increasing cellular uptake. To demonstrate that PRNPs could realize the selective reduction of OCs and synovial macrophages at sites of inflammation in AIA rats with advanced arthritis, we measured the level of cellular apoptosis in the inflamed joints, the number of OCs and macrophages in the synovial joints, and the cytokine profiles in blood and joints.

AIA rats with advanced arthritis were randomly divided into five groups based on their treatment with either saline, CEL solution, CEL-NPs, CEL-RNPs, or CEL-PRNPs. Saline or 1 mg/kg CEL equivalents of CEL solution, CEL-NPs, CEL-RNPs, or CEL-PRNPs were intravenously injected into rats every other 2 days. Ankle joints were collected for TUNEL staining two days after the last treatment. Inflammatory macrophages and OCs were determined by immunofluorescence analysis of CD68 and CD51 (red fluorescence). As shown in Fig. 7a and Supplementary Fig. 10, free CEL solution led to a very low level of apoptosis in inflamed joints, whereas CEL-PRNPs triggered the highest level

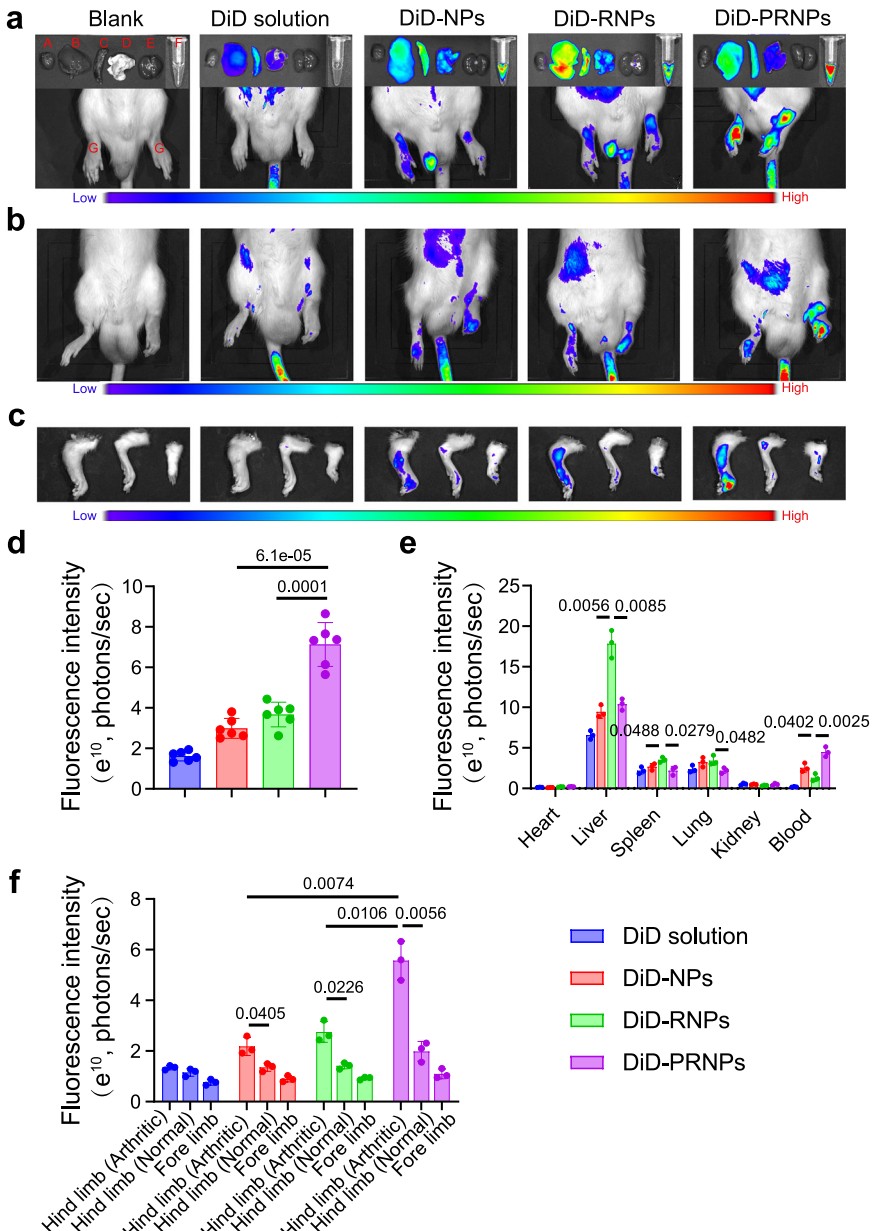

**Fig. 5 PRNPs selectively accumulate in inflamed joints of rats with advanced arthritis. a** Ex vivo DiD fluorescence images showing the biodistribution of NPs, RNPs, and PRNPs in AIA rats with advanced arthritis (A, heart; B, liver; C, spleen; D, lung; E, kidney; F, Blood; G, arthritic joint) at 24 h post injection. **b** In vivo DiD fluorescence images showing the arthritic joint distribution of free DiD, and DiD-loaded NPs, RNPs, and PRNPs in AIA rats with a unilateral inflamed joint at 24 h post injection. **c** Ex vivo DiD fluorescence images in the inflamed joints and un-inflamed joints from AIA rats with a unilateral inflamed joint at 24 h post injection with free DiD, and DiD-labeled NPs, RNPs, or PRNPs. **d** The statistical graphs of the fluorescence intensity of inflamed joints based on the semi-quantitative analysis of the ex vivo fluorescence images after *i.v.* administration of free DiD or DiD-labeled nanoparticles. Data represent mean ± SD ($n = 6$ inflamed joints from 3 independent animals). Statistical significance was determined by two-sided Student's *t* test. **e** The statistical graphs of the fluorescence intensity of major organs from AIA rats with advanced arthritis after i.v. administration of free DiD or DiD-labeled nanoparticles. Data represent mean ± SD ($n = 3$ independent animals). Statistical significance was determined by a two-sided Student's *t* test. **f** The statistical graphs of the fluorescence intensity of inflamed joints and un-inflamed joints from AIA rats with a unilateral inflamed joint after i.v. administration of free DiD or DiD-labeled nanoparticles. Data represent mean ± SD ($n = 3$ independent animals). Statistical significance was determined by a two-sided Student's *t* test. DiD 1,1'-dioctadecyl-3,3,3',3'-tetramethyl indodicarbocyanine, 4-chlorobenzenesulfonate salt, DiD-NPs DiD labeled poly (D, L-lactide-co-glycolide) (PLGA) nanoparticles, DiD-RNPs DiD-labeled RGD peptide-modified PLGA nanoparticles, DiD-PRNPs DiD-labeled matrix metalloproteinase 9 (MMP9)-cleavable polyethylene glycol (PEG)- and RGD peptide-modified PLGA nanoparticles.

of apoptosis in inflamed joints among all of the treatment groups. Furthermore, apoptotic cells induced by CEL-PRNPs were mainly inflammatory macrophages and OCs (Supplementary Fig. 11).

To determine the abundance of OCs and synovial macrophages in inflamed joints, ankle joints, and blood were collected for TRAP staining, immunohistochemical staining, and enzyme-linked immunosorbent assays (ELISA) 2 days after the last treatment. OCs located in the areas of bone erosion are stained red when using TRAP staining assay. TRAP staining results showed that CEL-PRNPs significantly reduced the number of OCs in ankle joints and resulted in negligible bone erosion (Fig. 7b). ELISA results revealed that the serum TRAP in the

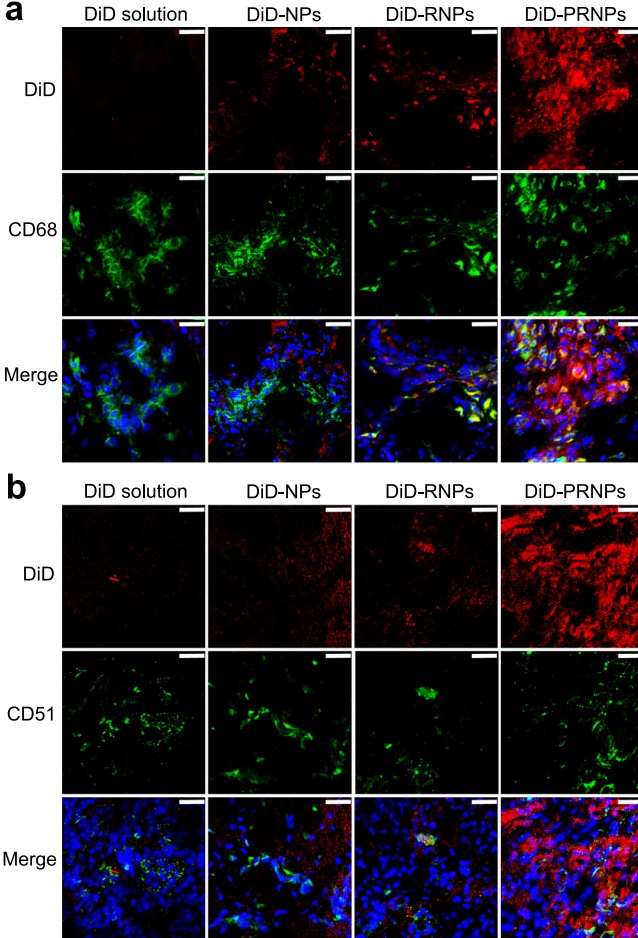

**Fig. 6 PRNPs are selectively distributed in OCs and inflammatory macrophages in arthritic joints of rats with advanced arthritis.** Confocal images showing the distribution of different DiD formulations in synovial macrophages (**a**) and OCs (**b**) in inflamed joints. Macrophages and OCs were determined by immunofluorescence analysis of CD68 and CD51 (green fluorescence), respectively. (Scale bar = 25 μm) (n = 3 independent animals). DiD 1,1′-dioctadecyl-3,3,3′,3′-tetramethyl indodicarbocyanine, 4-chlorobenzenesulfonate salt, DiD-NPs DiD labeled poly (D, L-lactide-co-glycolide) (PLGA) nanoparticles, DiD-RNPs DiD-labeled RGD peptide-modified PLGA nanoparticles, DiD-PRNPs DiD-labeled matrix metalloproteinase 9 (MMP9)-cleavable polyethylene glycol (PEG)- and RGD peptide-modified PLGA nanoparticles.

CEL-PRNPs group was comparable to that of the normal group (Supplementary Fig. 12). Immunohistochemical staining showed that, in comparison with free CEL, CEL-NPs, and CEL-RNPs treatment with CEL-PRNPs remarkably decreased the number of CD68+ macrophages in the synovium (Fig. 7b). These results suggested that CEL-PRNPs efficiently reduced the abundance of both OCs and inflammatory macrophages in AIA rats with advanced arthritis.

The superabundant presence of OCs in RA severely disrupts the balance of bone function[60,61]. To confirm the possibility that CEL-PRNPs could restore bone function balance by decreasing OCs abundance, we determined the expression levels of RANKL and osteoprotegerin (OPG) in arthritic joints. Previous studies have demonstrated that the RANKL-OPG system is important for regulating the balance between bone resorption and formation[62,63] and increased RANKL expression and a higher RANKL/OPG ratio may contribute toward the inefficient bone erosion repair in RA[64,65]. We observed that the CEL-PRNPs

group maintained a lower level of RANKL expression and a smaller RANKL/OPG ratio in the arthritic joints and blood than CEL, CEL-NPs, and CEL-RNPs groups, at a level comparable to that of the normal group (Fig. 7c; Supplementary Figs. 13 and 14), thereby demonstrating the recovery of bone function balance. In accordance with this, CEL-PRNPs promoted bone damage repair, as indicated by the significant accumulation of osteocalcin (OCN)-positive osteoblasts and the increased expression of alkaline phosphatase (ALP) in the arthritic joints (Fig. 7d; Supplementary Fig. 15).

Macrophages are the main producers of inflammatory cytokines including TNF and IL-1β in RA[66,67]. We found that the CEL-PRNPs group had the lowest expression levels of TNF and IL-1β in blood among all of the treatment groups (Fig. 7c). The immunohistochemical assays similarly revealed that CEL-PRNPs effectively reduced the secretion of TNF and IL-1β in ankle joints (Fig. 7d; Supplementary Fig. 15). Thus, CEL-PRNPs likely mediated the reduction of inflammatory macrophage abundance in arthritic joints. The above results demonstrated the marked ability of CEL-PRNPs in restoring bone function balance and reducing inflammatory cytokines secretion, through the reduction of the number of OCs and inflammatory macrophages in arthritic joints.

**CEL-PRNPs alleviate joint inflammation and bone erosion in rats.** Finally, the therapeutic efficacy of CEL-PRNPs treatment was evaluated in AIA rats with advanced arthritis. The AIA rat developed severe swelling in the ankles and paws after 17 days of arthritis induction. Saline or various CEL-loaded PLGA nanoparticles were intravenously injected into rats (dose of 1 mg/kg for CEL) (Fig. 8a). Anti-TNF, as one of the benchmarks in the treatment of RA, was also employed to treat AIA rats with advanced arthritis in this study. Free CEL and anti-TNF showed relatively low efficacy in decreasing the paw thickness and ankle diameters of AIA rats with advanced arthritis. In contrast, CEL-PRNPs showed higher efficacy in reducing swelling in ankle joints and paws compared with CEL-NPs and CEL-RNPs, yielding an ankle diameter and paw thickness closer to that of the normal group at the study endpoint (Fig. 8b, c). To further illustrate that CEL-PRNPs could control inflammation and reduce cartilage destructions, the ankle joints of rats were sectioned for histological analysis at the study endpoint. H&E-stained sections from the saline group showed severe synovial hyperplasia, along with bone and cartilage destruction. The free CEL group displayed a limited effect in reducing these symptoms, while CEL-NPs and CEL-RNPs reduced synovial inflammation and decreased the loss of cartilage to some extent compared with the saline group (Fig. 8d). However, safranin-O and toluidine blue staining revealed that the GAG levels in CEL-NPs and CEL-RNPs were lower than those in the control AIA rats (AIA rats following 17 days of arthritis induction) (Supplementary Fig. 1), suggesting that the cartilage damage was inefficiently reversed and remained progressive. In contrast, H&E results showed mild synovial hyperplasia in the CEL-PRNPs group. Additionally, the CEL-PRNPs group had larger positive areas for safranin-O and toluidine blue staining, which were closer to those of the normal group (Fig. 8d; Supplementary Fig. 16). These results demonstrated that CEL-PRNPs effectively alleviated synovial inflammation and reduced cartilage destructions in AIA rats with advanced arthritis.

We proved that AIA rats following 17 days of disease induction had obvious bone erosion and significant loss of BMD (Fig. 1). Using the micro-CT analysis, the inflamed ankle joints on day 29 after treatment were shown to exhibit rough bone surfaces and serious bone erosion in the saline group, with a significant

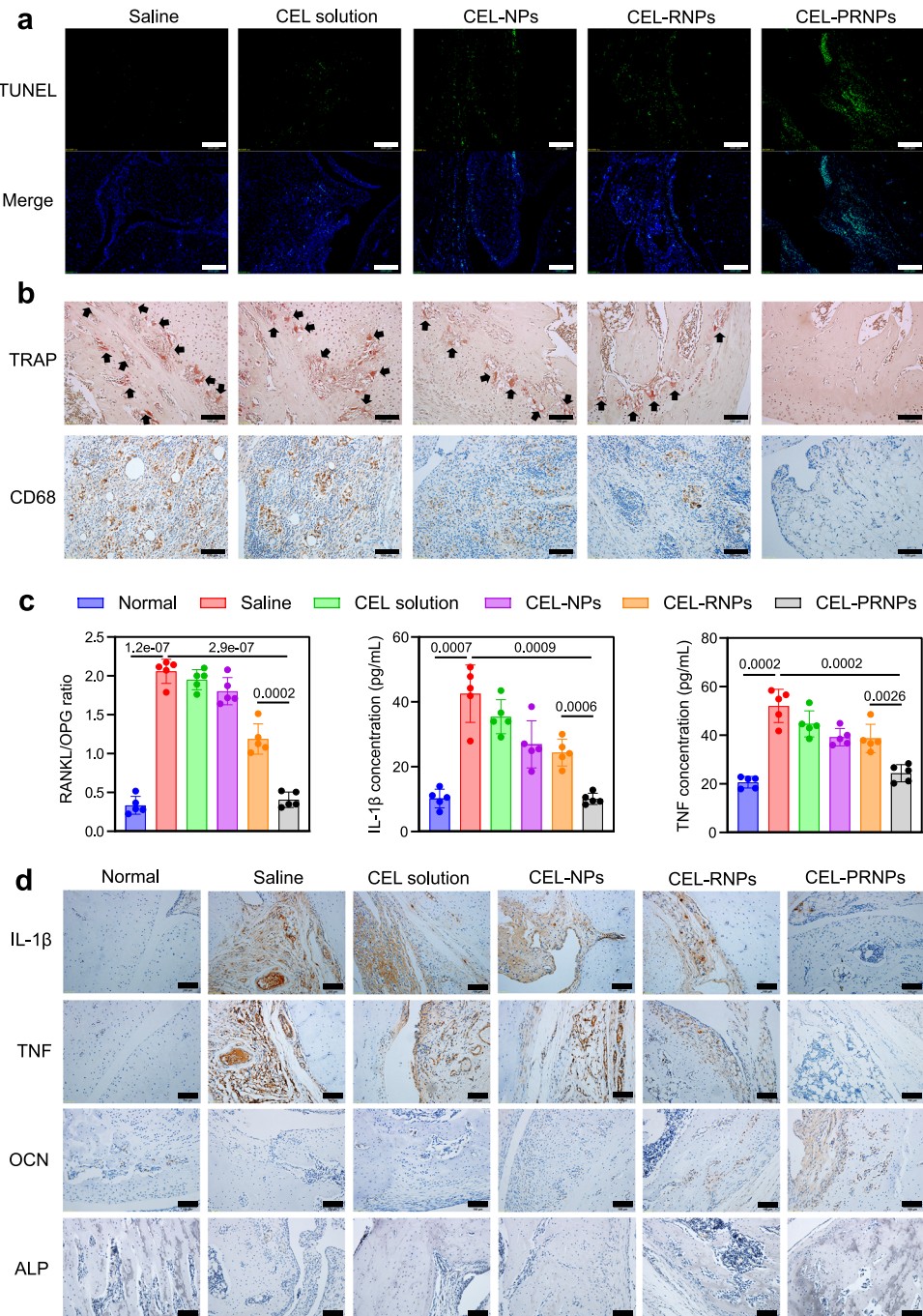

**Fig. 7 CEL-PRNPs reduce the number of OCs and inflammatory macrophages in joints of rats with advanced arthritis. a** TUNEL immunofluorescence staining in ankle joints from AIA rats receiving the indicated treatment (Scale bar = 200 μm) (n = 5 independent animals). **b** Immunohistochemical analyses of the TRAP-stained OCs and CD68-stained synovial macrophages in the joint tissues from rats receiving the indicated treatment (Scale bar = 100 μm) (n = 5 independent animals). **c** RANKL/OPG ratio in arthritic joints, IL-1β secretion in blood, and TNF secretion in blood from rats receiving the indicated treatment. Data represent mean ± SD (n = 5 independent animals). Statistical significance was determined by a two-sided Student's t test. **d** Detection of IL-1β, TNF, OCN, and ALP expression levels in arthritic joints in different groups. Arthritic joints in different groups were stained with IL-1β, TNF, and OCN antibodies, respectively. ALP was stained light–dark in arthritic joints from different groups (Scale bar = 100 μm) (n = 5 independent animals). CEL celastrol CEL-NPs CEL-loaded poly (D, L-lactide-co-glycolide) (PLGA) nanoparticles, CEL-RNPs CEL-loaded RGD peptide-modified PLGA nanoparticles, CEL-PRNPs CEL-loaded matrix metalloproteinase 9 (MMP9)-cleavable polyethylene glycol (PEG)- and RGD peptide-modified PLGA nanoparticles, TUNEL TdT-mediated dUTP nick end labeling, TRAP tartrate-resistant acid phosphatase, RANKL receptor of activator of NF-kB ligand, OPG osteoprotegerin, IL-1β interleukin-1 β, TNF tumor necrosis factor, OCN osteocalcin, ALP alkaline phosphatase.

reduction in BMD and an increase in BS/BV compared with the normal group. Free CEL showed moderate efficacy in reducing bone erosion. Treatment with anti-TNF, CEL-NPs, and CEL-RNPs increased the BMD compared with free CEL but failed to repair the bone damage. Remarkably, the CEL-PRNPs group showed smooth bone surfaces and had a high BMD, closer to those of the normal group (Fig. 9a–d), demonstrating that CEL-PRNPs efficiently reversed bone erosion. The trabecular

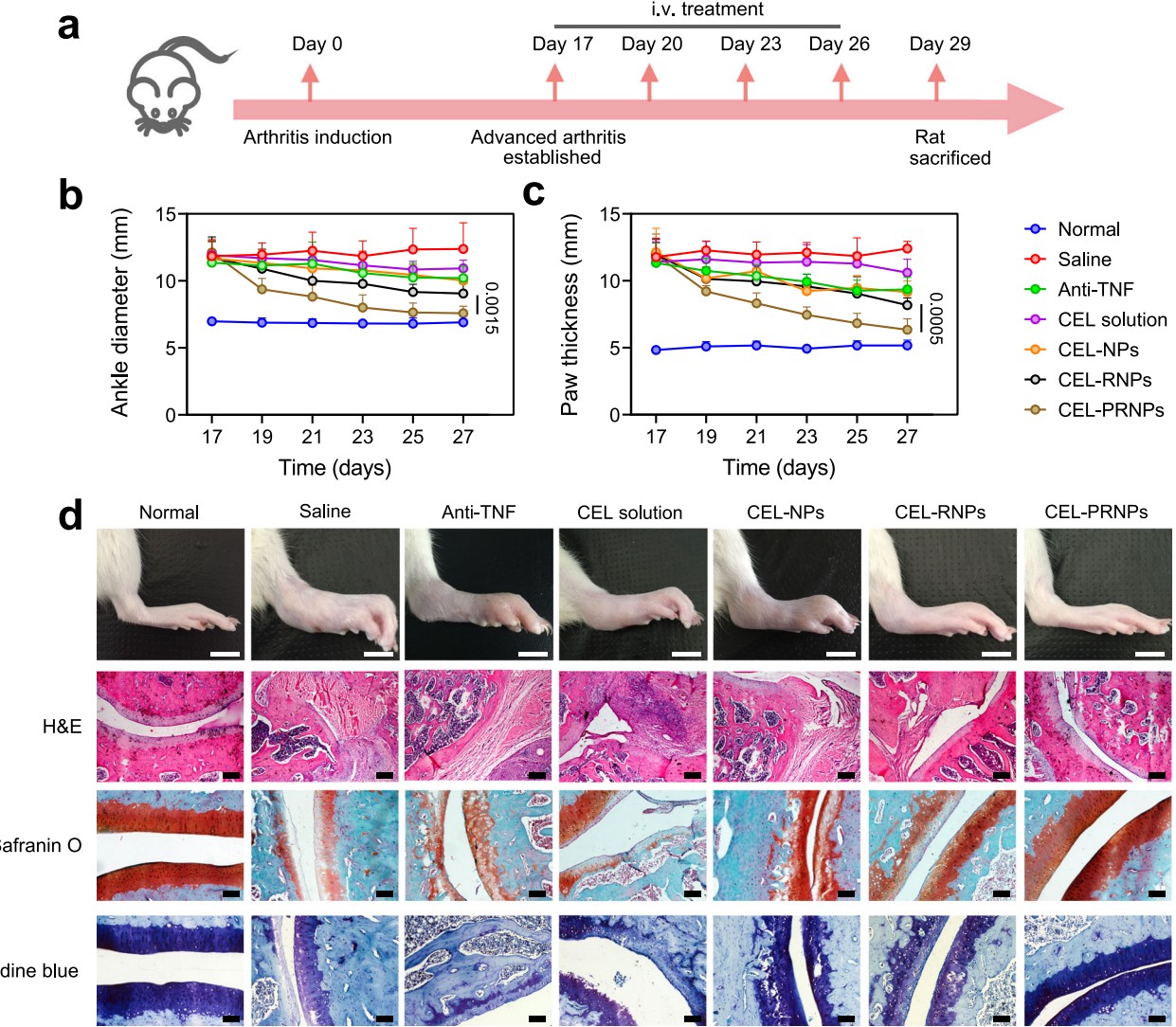

**Fig. 8 Therapeutic efficacy of CEL-PRNPs in rats with advanced arthritis. a** The schematic illustration of CEL-PRNPs treatment. **b, c** Ankle diameter (**b**) and paw thickness (**c**) of AIA rats were recorded every other day during the treatment period. Data represent mean ± SD (n = 7 independent animals). Statistical significance was determined by a two-sided Student's t test. **d** Representative photographs of hindlimbs at the endpoint of the experiment from different treatment groups (Scale bar = 10 mm); histopathology evaluation of ankle joints was identified using H&E (scale bar = 200 μm), safranin-O and toluidine blue staining (scale bar = 100 μm) (n = 5 independent animals). i.v. intravenous, anti-TNF anti-TNF (tumor necrosis factor) antibody, CEL celastrol, CEL-NPs CEL-loaded poly (D, L-lactide-co-glycolide) (PLGA) nanoparticles, CEL-RNPs CEL-loaded RGD peptide-modified PLGA nanoparticles, CEL-PRNPs CEL-loaded matrix metalloproteinase 9 (MMP9)-cleavable polyethylene glycol (PEG)- and RGD peptide-modified PLGA nanoparticles, H&E hematoxylin-eosin.

parameters also confirmed that CEL-PRNPs treatment was the most efficient in increasing the trabecular number (Tb.N) and trabecular bone thickness (Tb.Th) while decreasing trabecular separation (Tb.Sp) among all of the treatment groups (Fig. 9e–g). Thus, these findings proved that CEL-PRNPs could effectively terminate the progression of bone damages and simultaneously repair bone erosion in a rat model of advanced RA.

## Discussion

Apoptosis is the process of natural programmed cell death used to maintain organism homeostasis[68]. For example, macrophages are the first line of defense against pathogens[69,70]. They are regulated by apoptosis to maintain immune homeostasis and protect the host against damage from excessive inflammation[21,22]. OCs are terminally differentiated cells and known for their unique function in promoting bone resorption and the formation of

resorption lacunae[14–16]. The natural apoptosis of OCs helps to balance body bone-remodeling homeostasis to prevent OCs-mediated bone damage[23,54]. However, macrophages and OCs are abnormally accumulated in RA joints[10,11,66]. Furthermore, macrophages and OCs in arthritic joints are reported to exhibit the decreased apoptotic rates compared with those from healthy controls[21,25]. Therefore, we proposed that enhancing the apoptosis of macrophages and OCs would be a promising treatment strategy for restoring immune homeostasis and bone function balance in arthritic joints.

Our previous study found that CEL could trigger the apoptosis of mesangial cells in glomerulonephritis[38]. In the present study, we demonstrated the CEL-induced apoptosis of pathogenic macrophages and OCs (Figs. 3b and 4e, f). However, CEL treatment shows severe toxicity in normal organs due to its off-targeting activity[38,71]. Indeed, significant CEL toxicity in the

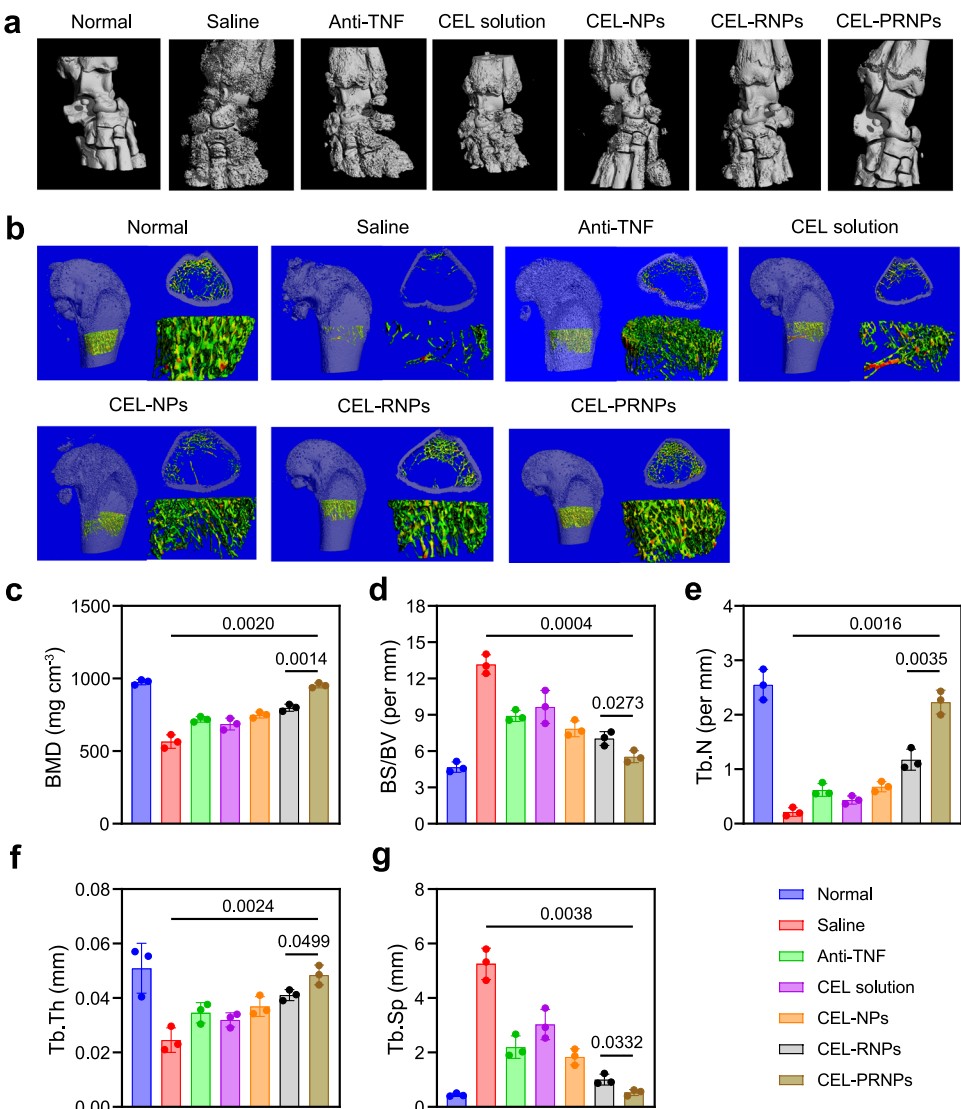

**Fig. 9 CEL-PRNPs reverse bone erosion in rats with advanced arthritis. a** Representative micro-CT images of the ankle joints at the endpoint of the experiment from different treatment groups in therapeutic efficacy study (*n* = 3 independent animals). **b** Representative micro-CT images of the trabecular in bone and the reconstructed trabecular structure (*n* = 3 independent animals). **c**, **d** Quantitative micro-CT analysis of BMD (**c**) and BS/BV (**d**) of the ankle joints at the endpoint of the experiment. Data represent mean ± SD (*n* = 3 independent animals). Statistical significance was determined by a two-sided Student's *t* test. **e**–**g** Quantitative micro-CT analysis of trabecular number (Tb.N) (**e**), trabecular bone thickness (Tb.Th) (**f**), and decreasing trabecular separation (Tb.Sp) (**g**). Data represent mean ± SD (*n* = 3 independent animals). Statistical significance was determined by a two-sided Student's *t* test. Anti-TNF anti-TNF (tumor necrosis factor) antibody, CEL celastrol, CEL-NPs CEL-loaded poly (D, L-lactide-co-glycolide) (PLGA) nanoparticles, CEL-RNPs CEL-loaded RGD peptide-modified PLGA nanoparticles, CEL-PRNPs CEL-loaded matrix metalloproteinase 9 (MMP9)-cleavable polyethylene glycol (PEG)- and RGD peptide-modified PLGA nanoparticles.

heart, liver, and brain was also observed in our study (Supplementary Figs. 17 and 18). Thus, it is necessary to specifically induce the apoptosis of inflammatory macrophages and OCs in arthritic joints to treat RA. To realize cell-specific drug delivery in the treatment of RA, the drug delivery system should be able to target both the inflammatory sites and the relevant cells within these sites. The RGD peptide is a well-known ligand of the integrins that are highly expressed on the surfaces of pathogenic macrophages and OCs in RA synovium[19,20]. Thus, we observed significant uptake of RGD-modified NPs (RNPs) by pathogenic macrophages (LPS-activated murine macrophages and human synovial macrophages) and OCs (Figs. 3a and 4a). However, integrins are overexpressed in many organs such as the liver[59], and this led to the enhanced distribution of RNPs in the liver and

higher CEL toxicity in this organ (Fig. 5a; Supplementary Figs. 17 and 18). Furthermore, integrins are also highly expressed on the neovascularization endothelial cells within the inflammatory sites of RA, which could lead to the selective distribution of RGD modified nanoparticles in endothelial cells[72,73]. This selective uptake by endothelial cells would hinder these nanoparticles from reaching the inflammatory microenvironment in RA. Our results also revealed that RNPs and PRNPs (in the presence of MMP9) exhibited high uptake by TNF-activated HUVECs, which mimic neovascularization endothelial cells. However, as a result of their MMP9-responsive PEG chain modification, PRNPs showed decreased distribution in activated HUVECs (Supplementary Fig. 19). Therefore, PRNPs with their inflammatory microenvironment-responsive properties offer not only good

in vivo safety but also enable the selective delivery of CEL-RNPs to the macrophages and OCs within the inflammatory micro-environment of arthritis.

Macrophages in the RA synovium produced cytokines, such as TNF and IL-1, to promote the progression of inflammation[66,67]. Thus, anti-cytokines are often applied in the clinical treatment of RA. In our rat RA model, we also found that anti-TNF treatment could significantly suppress the inflammation in early arthritis (Supplementary Fig. 20). However, the anti-TNF treatment showed reduced efficacy and limited bone protection in advanced arthritis (Figs. 8 and 9). What's more, anti-TNF and anti-IL-1 therapies are clinically effective in only 40% of patients, and the disease recurs when treatment is stopped[21,30,32]. This might be due to synovial macrophages producing other inflammatory mediators besides TNF and IL-1, such as IL-6 and IL-15[66,74]. In addition, synovial macrophages have a prolonged life span due to their insufficient apoptosis[21,75]. In contrast with anti-TNF treatment, CEL-loaded nanoparticles can target the synovial macrophages as a whole to induce their apoptosis. Furthermore, our CEL-PRNPs could selectively deliver CEL to inflammatory macrophages and OCs in arthritic joints to effectively induce the apoptosis of both cell types, thus significantly suppressing inflammation and terminating bone erosion (Figs. 8 and 9). Of note, the effective OCs depletion induced by CEL-PRNPs efficiently restored the balance of bone function, as indicated by the reduced RANKL expression and decreased RANKL/OPG ratio (Fig. 7c; Supplementary Figs. 13 and 14). The recovery of bone function balance induced by CEL-PRNPs promoted bone erosion repair by increasing ALP expression and the accumulation of OCN-positive osteoblasts (Fig. 7d).

Secondary osteoporosis occurs widely among RA patients because the increased number of OCs in RA severely disrupts the OC–osteoblast axis and enhances the bone resorption function of OCs[12,76]. Furthermore, glucocorticoids, one of the most common anti-inflammatory agents, are frequently applied in the treatment of patients with RA. Unfortunately, one severe side effect of glucocorticoids is osteoporosis induced by the prevention of calcium absorption[29]. Interestingly, the dual-targeting scaffold CEL-PRNPs developed in this study could effectively increase the bone density, trabecular number, and bone thickness, and decrease the trabecular separation, thereby demonstrating the efficient control of osteoporosis in advanced arthritis (Fig. 9). Such phenomena were mainly attributed to the enhanced OCs apoptosis induced by CEL-PRNPs effectively restoring the bone function balance. Therefore, our findings might suggest a new direction in nanomedicine in the osteoporosis control of RA patients.

PLGA, PEG, and RGD have been approved in clinic application and the substrate peptide of MMP9 also consists of essential amino acids for humans. CEL-PRNPs treatment showed good efficacy and led to negligible off-target apoptosis in rats. But further clinical translation, long-term toxicity of using PRNPs should be thoroughly evaluated in the future. In addition, knockout mice (αvβ3 integrin KO mice and MMP9$^{-/-}$ mice) could be adopted to evaluate the efficacy of CEL-PRNPs. The in vivo studies in knockout mice will be advantageous to the clinical translation of this developed drug delivery platform.

In summary, we developed inflammatory macrophages and OCs dual-targeted strategy based on MMP9-responsive CEL-PRNPs for advanced inflammatory arthritis treatment. CEL-PRNPs were shown to efficiently target both inflammatory macrophages and OCs after responding to MMP9 in the inflammatory microenvironment of arthritis. Targeting of inflammatory macrophages promoted their apoptosis, thereby reducing inflammation in arthritic joints. Targeting of OCs promoted OC apoptosis and inhibited their osteoclastic function,

consequently restoring the balance of bone function. Accordingly, CEL-PRNPs efficiently controlled joint inflammation, reversed bone erosion, and prevented secondary osteoporosis. Taken together, CEL-PRNPs realized the targeting of both synovial inflammation and bone erosion in advanced arthritis. This strategy for the selective apoptosis of inflammatory macrophages and OCs in arthritic joints shows great promise in inflammatory remission, bone erosion repair, and secondary osteoporosis prevention in advanced inflammatory arthritis.

## Methods

**Materials**. CEL (Catalog # A0106) was obtained from Chengdu Must Biotechnology (Chengdu, China). Poly (ethylene glycol)-poly (D, L-lactide-co-glycolide 50/50) (PEG$_{2000}$-PLGA$_{20000}$) and maleimide-PEG$_{2000}$-PLGA$_{20000}$ (Mal-PEG$_{2000}$-PLGA$_{20000}$) were obtained from the University of Electronic Science and Technology of China (Chengdu, China). Cys-RGD peptide and mPEG$_{2000}$-MMP9 cleavable Cys-peptide (mPEG$_{2000}$-GPLGLAGQC) were custom-synthesized by GL Biochem (Shanghai, China). RANKL (Catalog # 315-11) and M-CSF (Catalog # 315-02) were obtained from PeproTech (Rocky Hill, USA). 1,1′-dioctadecyl-3,3,3′,3′-tetramethyl indodicarbocyanine, 4-chlorobenzenesulfonate salt (DiD) (Catalog # M9379) was obtained from ChemBridge (San Diego, USA). LPS (Catalog # L4391) and human active MMP9 protein (Catalog # PF024) were purchased from Sigma Aldrich Co., LLC. (St. Louis, USA). Annexin V-FITC Apoptosis/Propidium Iodide Detection Kit (Catalog # G003-1-3) and TRAP Stain Kit (Catalog # D023-1-1) were from Nanjing Keygen Biotech. Co., Ltd. (Nanjing, China). JC-1 molecular probe (Catalog # C2005) was acquired from Beyotime Institute of Biotechnology (Haimen, China). In Situ Cell Death Detection Kit (Catalog # 11684795910) for TUNEL assay was from Roche (Basel, Switzerland). ELISA kits to assay the levels of rat RANKL (Catalog # SBJ-R0631) and OPG (Catalog # SBJ-R0631) were obtained from Nanjing Senbeijia Biological Technology Co., Ltd. (Nanjing, China). ELISA kits to assay the levels of rat TNF (Catalog # KRC3011) and IL-1β (Catalog # BMS630) were purchased from Invitrogen (Shanghai, China).

**Patient samples**. Peripheral blood samples and synovial tissues were obtained from three female patients with late-stage RA (according to the American College of Rheumatology criteria) undergoing joint replacement surgery. All samples were collected from Xiangya Hospital of Central South University. Informed consent was obtained from all patients, and ethical approval was obtained from the Ethics Committee of the Xiangya Hospital of Central South University (approval No. 2019010305).

**Animals**. Healthy male Wistar rats (200 ± 20 g, 5 weeks old) and male C57BL/J mice (20 ± 2 g, 6 weeks old) were obtained from Chengdu Dashuo Experimental Animal Co., Ltd. (Chengdu, China). Animals were housed in specific pathogen-free conditions at a standard temperature of 22 ± 2 °C and a relative humidity of 55% (45–70%) in a 12:12-h light–dark cycle. All animal studies were conducted according to the requirements of the national act regarding the use of experimental animals (China) and complied with the guidelines evaluated and approved by the Animal Ethics Committee of Sichuan University.

**Cells**. The in vitro OC differentiation was carried out as previously described[77]. In brief, bone marrow cells were isolated from the tibiae of C57BL/J mice and these cells were cultured with 30 ng/mL of M-CSF for 2 days and used as BMMs. To generate OCs, the BMMs were cultured in the presence of 100 ng /mL of RANKL and 30 ng/mL of M-CSF for 4 days. To obtain LPS-activated macrophages, BMMs were treated with 10 ng/mL of LPS for 48 h. All cells were cultured in RPMI-1640 medium containing 10% FBS and 100 U/mL of penicillin–streptomycin under 5% CO$_2$ at 37 °C.

Synovial tissue specimens were gained from patients with late-stage RA undergoing joint replacement surgery. The gained tissue specimens were washed, cut into small pieces and digested with collagenase. Tissue debris was removed by forcing the sample through a 70-μm cell strainer, thereby producing a cell suspension. Magnetic-activated cell sorting method[78,79] was adopted to isolate the macrophages from synovium-derived cells. Synovial macrophages were isolated to a high percentage of purity (>95%) with the use of MACS CD14 MicroBeads (Miltenyi Biotech, Germany). Primary synovial macrophages were cultured in Dulbecco's Modified Eagle's Medium (DMEM) containing 10% FBS and 100 U/mL of penicillin–streptomycin under 5% CO$_2$ at 37 °C.

Blood samples were obtained from patients diagnosed with late-stage RA. Peripheral blood mononuclear cells (PBMCs) were isolated using Ficoll–Paque (Miltenyi Biotech) density gradient centrifugation according to the manufacturer's instructions. PBMCs were cultured in the presence of 100 ng/mL of RANKL and 30 ng/mL of M-CSF for 4 days to generate OCs. RA patients-derived OCs were cultured in RPMI-1640 medium containing 10% FBS and 100 U/mL of penicillin–streptomycin under 5% CO$_2$ at 37 °C.

Human umbilical vein endothelial cells (HUVECs) were purchased from the Chinese Academy of Sciences Cell Bank for Type Culture Collection (Shanghai, China). HUVECs were cultured in RPMI-1640 medium containing 10% FBS and 100 U/mL of penicillin–streptomycin under 5% $CO_2$ at 37 °C. To activate HUVECs, cells were treated with cells with 50 ng/mL of TNF for 24 h.

**Synthesis and characterization of RGD-PEG$_{2000}$-PLGA$_{20000}$.** RGD-PEG$_{2000}$-PLGA$_{20000}$ was synthesized *via* the maleimide-thiol coupling reaction. Briefly, Mal-PEG$_{2000}$-PLGA$_{20000}$ and Cys-RGD (molar ratio = 1:2) were reacted in a solvent mixture comprising chloroform/MeOH (v/v = 2:1) with gentle stirring at room temperature for 12 h. The solvent was evaporated in a vacuum and the residue was re-dissolved with chloroform. The insoluble material (unreacted Cys-RGD) was filtered out, and the filtrate was evaporated in a vacuum to obtain RGD-PEG$_{2000}$-PLGA$_{20000}$. The successful synthesis of RGD-PEG$_{2000}$-PLGA$_{20000}$ was confirmed by $^1$H-NMR.

**Preparation and characterization of CEL-NPs, CEL-RNPs, and CEL-PRNPs.** CEL-NPs were prepared using an emulsion/solvent evaporation method. Briefly, PEG$_{2000}$-PLGA$_{20000}$ and CEL were dissolved in chloroform to form the oil phase. The resultant oil phase was then added into an aqueous phase. The mixture was emulsified by sonication with a probe sonicator (Ningbo Xinzhi Biotechnology Co. Ltd.; Ningbo, China). CEL-NPs were then prepared after the chloroform was evaporated at low pressure. CEL-RNPs were prepared as indicated above using PEG$_{2000}$-PLGA$_{20000}$, MAL-PEG$_{2000}$-PLGA$_{20000}$, and RGD-PEG$_{2000}$-PLGA$_{20000}$, and CEL was dissolved in chloroform to form the oil phase. To obtain CEL-PRNPs, mPEG$_{2000}$-GPLGLAGQC was conjugated with CEL-RNPs in PBS with pH 7.4 at room temperature for 4 h[52]. Unconjugated mPEG$_{2000}$-GPLGLAGQC was removed by elution through a Sephadex G-75 column.

The particle sizes and zeta potentials of CEL-NPs, CEL-RNPs, and CEL-PRNPs were determined by dynamic light scattering (DLS) using a Zetasizer Nano ZS90 instrument (Malvern Panalytical, Malvern, UK). The encapsulation efficiency (EE %) was determined by ultrafiltration method. The morphology of CEL-NPs, CEL-RNPs, and CEL-PRNPs was determined by TEM (H-600, Hitachi, Japan).

To study the serum stability, the prepared CEL-NPs, CEL-RNPs, and CEL-PRNPs were stored in 10% FBS at 37 °C for 24 h. Changes in particle size were determined by DLS.

**Cellular uptake study.** BMMs, HUVECs, activated HUVECs, OCs, and macrophages derived from mice and patients were seeded in 12-well plates and were treated with C6-NPs, C6-RNPs, and C6-PRNPs (with or without 5 μg/mL of MMP9) in the serum-free medium. After a 1 h incubation, cells were collected, centrifuged, and then suspended in phosphate-buffered saline (PBS). The fluorescence intensity of C6 was measured by a flow cytometer (BD FACSCelesta, USA).

For the qualitative analysis of cellular uptake, cells seeded in glass-bottomed dishes were treated as above. After incubation with C6-loaded NPs for 1 h, cells were fixed and stained with DAPI in the dark. The fluorescence images were then obtained using a laser scanning confocal microscope (Olympus FluoView FV 1000, USA).

**Apoptosis study.** OCs and macrophages were treated with CEL-NPs, CEL-RNPs, or CEL-PRNPs (the equivalent of 100 ng/mL of CEL and with or without 5 μg/mL of MMP9) for 24 h. Cell apoptosis was determined using an Annexin V-FITC Apoptosis/Propidium Iodide Detection Kit according to the manufacturer's instructions and the flow cytometric images were analyzed with FlowJo V10. The mitochondrial membrane potentials of these apoptotic macrophages were measured by incubating the macrophages with 5 μg/mL of JC-1 for 30 min. The fluorescent images were obtained using a laser scanning confocal microscope.

**TRAP assay.** To generate OCs, BMMs were cultured in the presence of 100 ng/mL of RANKL and 30 ng/mL of M-CSF for 4 days. The control group comprised BMMs cultured with M-CSF only for 4 days. At the study endpoint, cells were stained by TRAP Stain Kit and observed with a light microscope (Axiovert 40CFL, Germany).

**AIA model.** The AIA model was developed according a previous study but with slight modifications[80]. In brief, healthy male Wistar rats (200 ± 20 g) were subcutaneously injected with Freund's adjuvant (200 μL) containing 10 mg/mL of heat-killed mycobacteria (Chondrex, Washington DC, USA) into the base of their tails. Rats with 4 days of disease induction were stipulated as AIA rats with early-stage arthritis and an advanced arthritis rat model was successfully developed 17 days after disease induction. In addition, AIA rats with a unilateral inflamed joint were induced by the subcutaneous injection of Freund's adjuvant (200 μL) containing 10 mg/mL of heat-killed mycobacteria into the paw of the right hind limb.

**Biodistribution of PRNPs.** Arthritic rats with advanced arthritis were established and intravenously injected with free DiD, DiD-NPs, DiD-RNPs, or DiD-PRNPs via the tail vein. The biodistribution of DiD in the ankle joints was analyzed 2, 8, 24,

and 48 h after administration using in vivo imaging analysis of DiD fluorescence with a Caliper IVIS Lumina III In Vivo Imaging System (Perkin Elmer, USA). At the end of this experiment, rats were sacrificed, and their blood, organs (heart, liver, spleen, lung, and kidney), and ankle joints were collected for ex vivo imaging of DiD fluorescence.

AIA rats with a unilateral inflamed joint were intravenously injected with free DiD, DiD-NPs, DiD-RNPs, or DiD-PRNPs via the tail vein. The biodistribution of DiD in the ankle joints was analyzed 24 h after administration using in vivo imaging analysis of DiD fluorescence with a Caliper IVIS Lumina III In Vivo Imaging System (Perkin Elmer, USA). At the end of this experiment, rats were sacrificed, their hind limbs and forelimbs were collected for ex vivo imaging of DiD fluorescence.

**Immunofluorescence staining.** Arthritic rats with advanced arthritis were intravenously injected with free DiD, DiD-NPs, DiD-RNPs, or DiD-PRNPs via the tail vein. Ankle joints were collected to prepare sections 24 h after administration. The prepared sections of 10 μm thickness were stained with rat anti-CD68 (Abcam, Cat#ab125212, 1/500 dilution) and anti-CD51 (Abcam, Cat#ab179475, 1/500 dilution). DAPI was used for the nuclear stain. The fluorescent distributions in synovial joints were observed with a laser scanning confocal microscope (Leica TCS SP8 CARS, Germany).

**TUNEL staining.** Saline or various CEL-loaded PLGA nanoparticles were respectively intravenously injected into RA rats (dose of 1 mg/kg for CEL). Rats have received this treatment twice, and 2 days after the last treatment, the rats were sacrificed and their arthritic joints were collected. The arthritic joints were sectioned for TUNEL, anti-CD68 (Abcam, Cat#ab125212, 1/500 dilution), and anti-CD51 (Abcam, Cat#ab179475, 1/500 dilution) co-staining, then observed by a laser scanning confocal microscope and analyzed with Image J1.52.

**Cytokine assay.** AIA rats with advanced arthritis were established as indicated above. Saline or various CEL-loaded PLGA nanoparticles were respectively intravenously injected into rats (dose of 1 mg/kg for CEL). The rats received this treatment four times and blood samples were collected at the study endpoint. The blood samples were centrifuged at 2295 g for 15 min at 4 °C to obtain the serum. TNF, IL-1β, RANKL, and OPG in the serum were detected using ELISA kits.

**Therapeutic efficacy study.** AIA rats with early-stage or advanced arthritis were used in the therapeutic efficacy study. Saline or 1 mg/kg of CEL equivalents of CEL solution, CEL-NPs, CEL-RNPs, or CEL-PRNPs were intravenously injected into rats every 3 days. Anti-rat TNF (1.5 mg/kg) was used as the positive control and injected intraperitoneally into rats every other two days. Paw thickness and the dimensions of ankle joints were measured every other day during the treatment.

**Histology and immunohistochemical study.** Ankle joints collected before the treatment and 2 days after the last treatment were fixed in 4% paraformaldehyde. Fixed ankle joints were then decalcified by daily changes of a 15% (w/v) tetra-sodium ethylenediaminetetraacetic acid solution for 2 months. The decalcified joints were subsequently embedded in paraffin and then sectioned for H&E, safranin O, toluidine blue, TRAP, CD68, MMP9, RANKL, OPG, OCN, ALP, TNF, and IL-1β staining. Ankle joints from both the early- and late-stage of AIA rats were also obtained for H&E, safranin O, toluidine blue, TRAP, CD68, and MMP9 staining. These sections were observed by a light microscope and analyzed with Image J1.52. anti-CD68: Abcam, Cat#ab125212, 1/500 dilution; anti-MMP9: Abcam, Cat#ab76003, 1/1000 dilution; anti-RANKL: Abcam, Cat#ab239607, 1/100 dilution; anti-OPG: Abcam, Cat#ab203061, 1/200 dilution; anti-OCN: Abcam, Cat#ab13420, 1/200 dilution; anti-ALP: Abcam, Cat#ab224335, 1/200 dilution; anti-TNF: Abcam, Cat#ab220210, 1/100 dilution; anti-IL-1β: Abcam, Cat#ab205924, 1/200 dilution.

**Bone assessment and micro-CT analysis.** The ankle joints collected before the treatment and 2 days after the last treatment in the therapeutic study were fixed in 4% paraformaldehyde and scanned at 80 kV and 500 μA with a resolution of 15 μm by ex vivo micro-computed tomography (Micro-CT, VivaCT 80, SCANCO Medical AG, Switzerland). The dataset was then reconstructed to obtain the 3D images of the joints and trabecular in the distal femur. Bone morphometric parameters including BMD, bone surface vs. bone volume (BS/BV). trabecular number (Tb.N), trabecular bone thickness (Tb.Th), and trabecular separation (Tb.Sp) were also quantitatively analyzed. Ankle joints from both early and late stages AIA rats were also applied for micro-CT analysis.

**Safety evaluation.** To assess the in vivo safety of CEL-PRNPs, healthy male Wistar rats (200 ± 20 g) were intravenously administered with 1 mg/kg of CEL equivalents of CEL solution, CEL-NPs, CEL-RNPs, or CEL-PRNPs, respectively. An equal volume of saline was injected into the control rats. Rats were sacrificed 2 days after the last treatment, and the blood and major organs (heart, liver, spleen, lung, kidney, and brain) were collected for serum enzyme and histopathological analyses.

In addition, collected organs were sectioned for TUNEL staining and then analyzed with a laser scanning confocal microscope.

**Statistical analysis**. All quantitative parameters were presented as mean with standard deviation. For a two-group comparison, a Student's two-sided $t$ test was performed for the statistical analysis. For multiple comparisons, the data were analyzed using a two-way analysis of variance (ANOVA). A significant difference was considered when the $P$ value was less than 0.05.

**Reporting summary**. Further information on research design is available in the Nature Research Reporting Summary linked to this article.

## Data availability

All data are available in the Article, Supplementary Information files, or from the corresponding author upon reasonable request. The source data underlying Figs. 1b, c, 2d, 3d–g, 4b–f, 5d–f, 7c, 8b, c, and 9c–g and Supplementary Figs. 3a, b, 7b, 8b, 9b–f, 10, 12, 14a–c, 15a–d, 16, 18a–d, 19b, and 20b, c and Supplementary Table 1 are provided as a Source Data file. Source data are provided with this paper.

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

## Acknowledgements
This work was financially supported by the National Natural Science Foundation of China (No. 81872804) and Sichuan major science and technology project on bio-technology and medicine (2018SZDZX0018). In addition, the authors would like to thank Dr. Li Chen from Analytical & Testing Center Sichuan University for her help with micro-CT scanning and analysis.

## Author contributions
T.G. and G.L. conceived and planned the study. C.D. carried out the experiments, generated and analyzed data, created Figs. 2a and 8a and Supplementary Fig. 20a, and wrote the original paper. Q.Z., P.H., and K.H. helped with animal and cell studies. B.Z. collected human samples and helped with related experiments. X.S., T.G., and Z.Z. helped with paper editing.

## Competing interests
The authors declare no competing interests.
