## [Peer Review File · Nature Communications]

Reviewers' comments:

Reviewer #1 (Remarks to the Author):

This manuscript describes the development of a nanoparticle therapeutic (CEL-PRNP) capable of inducing apoptosis of macrophages and osteoclasts, which was used to treat rat AIA. The PRNP was designed to express RGD peptides to target $\alpha v\beta 3$ integrin positive cells and another protective PEG chain with a MMP9 cleavable peptide. When in an inflammatory environment where MMP9 is expressed, the protective molecule would be cleaved allowing the RGD peptide to bind the integrin positive cells, allowing the CEL to be internalized and released inducing apoptosis. The CEL-PRNP was effective at treating both early and late AIA, although the in vivo mechanism is less clear. An approach to target pathogenic macrophages may be helpful in a subset of patients with RA. There were a number of concerns with the manuscript as submitted, and addressing them may strengthen the final product.

- 1) The authors should have the manuscript reviewed by someone more facile with English. It was difficult to follow at times. This was especially true in the introduction, which has many misstatements and exaggerations. A couple of examples: synovial hyperplasia is not a symptom; continually suffer; extremely low; and various tumor cells.
- 2) While the therapeutic approach described may potentially be appropriate for some patients with RA, there are already many effective therapies that prevent joint erosion and destruction. Also recent studies have shown that JAK inhibitors may reverse erosions, as may other biologic therapies. The rationale should be more realistically presented.
- 3) It took a lot of effort to figure out what the different compounds were, and what the abbreviations represented. Figure 2A gives a clear description, but the manuscript text did not. RNP was not described in the text, that I could find. Expanding in the text and referring to the diagram would help. Also, Figure 2B is really the hypothesis being tested and is not actually data.
- 4) The last paragraph of the introduction is presented as what was expected, rather than what was observed.
- 5) There were NO data in the manuscript that addressed rheumatoid arthritis (RA). Rat AIA is not RA and should not be referred to as RA. It is an experimental model of RA.
- 6) The term activated macrophages was over used. In vitro differentiated macrophages treated with LPS would be appropriately referred to as activated. However, it is unlikely that every macrophage in an AIA joint is activated. It is likely that tissue-resident macrophages are also in the joint. The state of activation of synovial tissue macrophages was not studied.
- 7) The data concerning the uptake in off target organs in figure 4A needs to be quantitated and statistically interrogated. This is a major point for using the PRNPs.
- 8) An understanding of the mechanism of action in the joint tissues should be more rigorously addressed. What cells are taking up the DIDs? Macrophages, OCs, endothelial cells, other cells?
- 9) As RANKL/OPG were quantitated on joint tissue, so should the cytokines (Figure 4D).
- 10) Does treatment with CEL-PRNP promote apoptosis of cells in the joints and which cells? Flow cytometry would be one way to address this. Do macrophages and OCs undergo apoptosis in the joints? Are apoptotic bodies taken up by the remaining macrophages? Are suppressive cytokines such as TGF β or IL-10 released? Does CEL-PRNP work in vivo as hypothesized?
- 11) The data in Fig 5D should be quantitated, the statements are not convincingly supported by the pictures.
- 12) What is the mechanism for cleavage of the MMP9 peptide in the joint? Is MMP9 expressed? MMP9 was added to the in vitro activated macrophages. How do we know that the MMP9 target peptide is cleaved in the joint?
- 13) How much cell death or other changes occurs in the off target organs of rats treated with CEL-PRNP?
- 14) A control using PRNP, without CEL, should be included in the study. What is the in vivo effect of the nanoparticles?

Minor Concerns:

- 15) The sentence beginning on line 151 is confusing and should be rewritten.
- 16) Figure 2C is not mentioned in the text, although it probably would go on line 157.
- 17) Why were murine macrophages used for in vitro studies, rather than rat?

Reviewer #2 (Remarks to the Author):

This manuscript aims for delivering celastrol (CEL) to selectively induce apoptosis of both types of cells (osteoclasts and macrophages) in RA joints. While this is a novel work and nothing wrong with the science, it lacks the required impact to be published as it is in Nat Comm Journal. The work is mostly focused on in vivo studies using AIA rat model, but would be most impactful if validation is done using human cells obtained from patients with RA (e.g. ex vivo models). The fate of PLGA NPs should also be clarified by providing bioelimination and cell internalization (dynamic of endocytosis/exocytosis) data. In addition, some refinement is required with respect to the manuscript contents. Grammar revision should be once performed. Include scale bars in all figures. Some references are also incomplete. In brief, this manuscript requires major revision prior its acceptance for publication.

Reviewer #3 (Remarks to the Author):

This is a manuscript that focuses on a novel method of delivering celastrol for suppressing arthritis. The bulk of the manuscript has been published by other groups looking at AIA model and celastrol treatment of macrophages and osteoclasts. The main novelty is the delivery vehicle. However, to demonstrate the efficacy of the prodrug, the authors should use alpha v Beta 5 intern receptor KO mice and MMP9^{-/-} mice.

Reviewer #1 (Remarks to the Author):

This manuscript describes the development of a nanoparticle therapeutic (CEL-PRNP) capable of inducing apoptosis of macrophages and osteoclasts, which was used to treat rat AIA. The PRNP was designed to express RGD peptides to target $\alpha\text{v}\beta\text{3}$ integrin positive cells and another protective PEG chain with a MMP9 cleavable peptide. When in an inflammatory environment where MMP9 is expressed, the protective molecule would be cleaved allowing the RGD peptide to bind the integrin positive cells, allowing the CEL to be internalized and released inducing apoptosis. The CEL-PRNP was effective at treating both early and late AIA, although the in vivo mechanism is less clear. An approach to target pathogenic macrophages may be helpful in a subset of patients with RA. There were a number of concerns with the manuscript as submitted, and addressing them may strengthen the final product.

1) The authors should have the manuscript reviewed by someone more facile with English. It was difficult to follow at times. This was especially true in the introduction, which has many misstatements and exaggerations. A couple of examples: synovial hyperplasia is not a symptom; continually suffer; extremely low; and various tumor cells.

Response: Per the Reviewer's suggestion, we have rewritten the manuscript with the help of an English native speaker. In addition, we have proofread our manuscript and corrected the errors in our revised manuscript.

2) While the therapeutic approach described may potentially be appropriate for some patients with RA, there are already many effective therapies that prevent joint erosion and destruction. Also recent studies have shown that JAK inhibitors may reverse erosions, as may other biologic therapies. The rationale should be more realistically presented.

Response: We appreciate the Reviewer's suggestion and agree that JAK inhibitors may reverse the bone erosions in RA. Such information including the efficacy and

potential side effects of JAK inhibition (*Arthritis Rheum.*, 2012, 64:970-981. *Nat. Rev. Rheumatol.*, 2017, 13: 234-243) has been added into the “INTRODUCTION” section of the revised manuscript.

3) It took a lot of effort to figure out what the different compounds were, and what the abbreviations represented. Figure 2A gives a clear description, but the manuscript text did not. RNP was not described in the text, that I could fine. Expanding in the text and referring to the diagram would help. Also, Figure 2B is really the hypothesis being tested and is not actually data.

Response: Per the Reviewer’s suggestion, we have expanded the description for RNPs (RGD modified PLGA nanoparticles) and referred to the diagram (**Fig. 2a**) in the revised manuscript. In agreement with the Reviewer’s opinion, **Fig. 2B** is not actually data but is presented to illustrate the antiarthritic mechanism of CEL-PRNPs in the treatment of advanced RA. Thus, we have presented this figure as **Supplemental Fig. 1** in the “SUPPLEMENTARY INFORMATION” file.

4) The last paragraph of the introduction is presented as what was expected, rather than what was observed.

Response: Per the Reviewer’s suggestion, we have replaced the last paragraph of “INTRODUCTION” section to what was observed in the revised manuscript.

5) There were NO data in the manuscript that addressed rheumatoid arthritis (RA). Rat AIA is not RA and should not be referred to as RA. It is an experimental model of RA.

Response: In agreement with the Reviewer’s opinion, rat AIA is an experimental model of RA. We intended to develop CEL-PRNPs for addressing advanced RA. Per the Reviewer’s suggestion, we have carried out relevant experiments on osteoclasts (OCs) and activated macrophages derived from RA patients. We studied the distribution behavior and apoptosis-inducing ability of CEL-PRNPs in both types of cells. Results revealed that PRNPs could also increase drug distribution and cellular

apoptosis on both OCs and activated macrophages derived from RA patients after responding to MMP9. (Fig. 4) These observations have been added and presented in the revised manuscript.

6) The term activated macrophages was over used. In vitro differentiated macrophages treated with LPS would be appropriately referred to as activated. However, it is unlikely that every macrophage in an AIA joint is activated. It is likely that tissue-resident macrophages are also in the joint. The state of activation of synovial tissue macrophages was not studied.

Response: In agreement with the Reviewer's opinion, there are macrophages including tissue-resident macrophages remaining quiescent in arthritic joints. However, previous studies found the significant increase of inflammatory macrophages in synovial tissues of RA patients compared with those of healthy controls, and these infiltrated macrophages were highly activated. (*Ann. Rheum. Dis.*, 1999, 58:648-652; *Adv. Drug. Deliv. Rev.*, 2004, 56:1205-1217; *Nat. Rev. Rheumatol.*, 2016, 12: 472-485) In addition, the degree of synovial macrophage infiltration correlates with the degree of disease severity. (*J. Rheumatol.*, 1998, 25:214-220; *Arthritis Rheum.*, 2003, 48: 339-347; *Nat. Rev. Rheumatol.*, 2016, 12: 472-485) Furthermore, CEL-PRNPs could selectively deliver drugs into activated macrophages (Fig. 3) rather than non-activated macrophages (BMMs). (Supplemental Fig. 8) Therefore, we didn't evaluate the state of synovial tissue macrophages in this study.

Supplemental Fig. 8 (a) Confocal images showing the cellular uptake of C6-loaded NPs, RNP or PRNP on BMMs. (b) Quantitative analysis of the cellular uptake of C6-loaded NPs, C6-loaded RNP, C6-loaded PRNP, or C6-loaded PRNP/MMP9 on BMMs.

RNPs or C6 loaded PRNPs on BMMs. Data represent mean \pm SD ($n = 3$).

7) The data concerning the uptake in off target organs in figure 4A needs to be quantitated and statistically interrogated. This is a major point for using the PRNPs.

Response: Per the Reviewer's suggestion, we have quantitated and statistically analyzed the drug distribution of various DiD formulations in major organs (heart, liver, spleen, lung and kidney) and inflamed joints. (**Fig. 5, Supplemental Fig. 9 and Supplemental Fig. 10**) In addition, we have evaluated the joint distribution of various DiD formulations in AIA rats with a unilateral inflamed joint. Results revealed that PRNPs selectively delivered drugs into inflamed joint rather than normal joint. (**Fig. 5b, c and f**)

Fig. 5 (a) *Ex vivo* DiD fluorescence images showing the biodistribution of NPs, RNPs and PRNPs

in AIA rats with advanced arthritis (A, heart; B, liver; C, spleen; D, lung; E, kidney; F, Blood; G, arthritic joint) at 24 h post-injection. (b) *In vivo* DiD fluorescence images showing the arthritic joint distribution of free DiD, and DiD-loaded NPs, RNPs and PRNPs in AIA rats with a unilateral inflamed joint at 24 h post-injection. (c) *Ex vivo* DiD fluorescence images in the inflamed joints and un-inflamed joints from AIA rats with a unilateral inflamed joint at 24 h post-injection with free DiD, and DiD-labeled NPs, RNPs or PRNPs. (d, e) The statistical graphs of the fluorescence intensity of inflamed joints (d) and major organs (e) based on the semi-quantitative analysis of the *ex vivo* fluorescence images after *i.v.* administration of free DiD or DiD-labeled nanoparticles. Data represent mean \pm SD ($n = 3$), $*P < 0.05$, $**P < 0.01$, $***P < 0.001$. (f) The statistical graphs of the fluorescence intensity of inflamed joints and un-inflamed joints from AIA rats with a unilateral inflamed joint after *i.v.* administration of free DiD or DiD-labeled nanoparticles. Data represent mean \pm SD ($n = 3$), $*P < 0.05$, $**P < 0.01$.

8) An understanding of the mechanism of action in the joint tissues should be more rigorously addressed. What cells are taking up the DIDs? Macrophages, OCs, endothelial cells, other cells?

Response: We appreciate the Reviewer's suggestion. To investigate whether PRNPs could target both OCs and activated macrophages in inflamed joints, the *in vivo* distribution behavior of different DiD formulations in both type of cells has been determined using immunofluorescent staining method. Activated macrophages and OCs in inflamed joints were determined by immunofluorescence analysis of CD68 and CD51, respectively. As shown in **Fig. 6**, the DiD fluorescence distribution of PRNPs in synovial joint was the highest among the three nanoparticle types, which was consistent with the results of *in vivo* and *ex vivo* imaging studies. In addition, free DiD and DiD-labeled NPs showed low levels of colocalization of the red (DiD) and green fluorescence (CD68 and CD51), suggesting the nonspecific distributions in inflamed joints. Whereas, the DiD fluorescence of RNPs and PRNPs was mainly overlapped with the green fluorescence in synovial joints. The DiD fluorescence of PRNPs group was significantly brighter than that of RNPs group. These results have demonstrated that PRNPs efficiently delivered drugs into OCs and activated

macrophages in inflamed joints.

Also, we have conducted *in vitro* experiments to investigate the distribution behaviors of PRNPs in human umbilical vein endothelial cells (HUVECs) and TNF- α -activated HUVECs (mimicking neovascularization endothelial cells). In HUVECs, all prepared nanoparticles displayed low level of drug distribution, demonstrating the low affinity of prepared nanoparticles to normal endothelial cells. **(Fig. 4a and d)** In activated HUVECs, RNPs and PRNPs (in the presence of MMP9) exhibited relatively high uptake by cells. However, as a result of MMP9-responsive PEG chains modification, PRNPs showed decreased distribution in activated HUVECs. **(Supplementary Fig. 19)** In addition, as shown in **Fig. 6**, the DiD fluorescence of PRNPs showed the highest drug distribution in synovial tissues among treated groups. These results have demonstrated that the MMP9-responsive PEG chains modification of PRNPs decreased RGD-mediated endocytosis in activated HUVECs. Consequently, PRNPs could extravasate from the blood vessels into the synovial tissues in arthritic joints and then targeting OCs and activated macrophages in inflammatory microenvironment (where MMP9 highly expressed).

Fig. 6 Confocal images showing the distribution of different DiD formulations in synovial macrophages (a) and OCs (b) in inflamed joints. Macrophages and OCs were determined by immunofluorescence analysis of CD68 and CD51 (green fluorescence), respectively. (Scale bar = 25 μ m) ($n = 3$).

Supplemental Fig. 19 (a) Confocal images (magnification 400 \times) of the cellular uptake of C6-loaded NPs, C6-loaded RNPs, and C6-loaded PRNPs in TNF- α -activated HUVECs. (b) Quantitative analysis of the cellular uptake of C6-NPs, C6-RNPs and C6-PRNPs in TNF- α -activated HUVECs after 1 h incubation at the C6 concentration of 50 ng/mL. *** $P < 0.001$.

Data represent mean \pm SD ($n = 3$).

9) As RANKL/OPG were quantitated on joint tissue, so should the cytokines (Figure 4D).

Response: Per the Reviewer's suggestion, we have quantitated the cytokine levels on joint tissues and the results have been shown in **Supplemental Fig. 15**. In addition, the serum levels of RANKL and OPG have been also measured, which were shown in **Supplemental Fig. 14**.

Supplemental Fig. 15 The relative quantity of IL-1 β (a), TNF- α (b), OCN (c) and ALP (d) in arthritic joints in different groups. ** $P < 0.01$, *** $P < 0.001$. Data represent mean \pm SD ($n = 5$).

10) Does treatment with CEL-PRNP promote apoptosis of cells in the joints and which cells? Flow cytometry would be one way to address this. Do macrophages and OCs undergo apoptosis in the joints? Are apoptotic bodies taken up by the remaining macrophages? Are suppressive cytokines such as TGF- β or IL-10 released? Does CEL-PRNP work in vivo as hypothesized?

Response: Thanks for the Reviewer's suggestion. We have applied TUNEL immunostaining method to evaluate the apoptosis in inflamed joints of AIA rats with advanced arthritis after various CEL formulation treatments. As shown in **Fig. 7a**,

CEL-PRNPs group showed the highest green fluorescent signal in arthritic joints, demonstrating the highest level of cellular apoptosis among all the treated groups. Furthermore, PRNPs could selectively deliver drugs to synovial macrophages and OCs in inflamed joints. **(Fig. 6)** CD68 and TRAP staining results also revealed that CEL-PRNPs was the most efficient in reducing the abundance of both synovial macrophages and OCs in inflamed joints. **(Fig. 7b)** Accordingly, CEL-PRNPs effectively induced apoptosis of synovial macrophages and OCs in arthritic joints of AIA rats with advanced arthritis.

In this study, we aimed to selectively inducing apoptosis of synovial macrophages and OCs in arthritic joints, thus reprogramming the inflammatory microenvironment and restoring the bone function balance, in an effort to control joint inflammation and reverse bone erosions in advanced RA. As shown in **Fig. 7**, we demonstrated that CEL-PRNPs effectively reduced the abundance of both types of cells *via* inducing increased apoptosis. The reduction of synovial macrophages resulted in the decrease of inflammatory cytokines (TNF- α and IL-1 β) secretion in both serum and arthritic joints, **(Fig 7c, d and Supplemental Fig. 15)** thus reducing swelling in ankle joints and paws. **(Fig 8b-d)** The reduction of OCs resulted in the recovery of bone function balance (decreased RANKL/OPG ratio), **(Fig 7c and Supplemental Fig. 14)** thus decreasing the loss of cartilage and repairing bone erosions. **(Fig. 9)** These results and observations have illustrated the antiarthritic mechanism of CEL-PRNPs *in vivo* and demonstrated that CEL-PRNP worked *in vivo* as we proposed.

Fig. 7 (a) TUNEL immunofluorescence staining in ankle joints from AIA rats receiving the indicated treatment (Scale bar = 200 μ m) ($n = 5$). (b) Immunohistochemical analyses of the TRAP stained OCs and CD68 synovial macrophages in the joint tissues from rats receiving the indicated treatment (Scale bar = 100 μ m) ($n = 5$). (c) RANKL/OPG ratio in arthritic joints, IL-1 β secretion in blood, and TNF- α secretion in blood from rats receiving the indicated treatment. $**P < 0.01$, $***P < 0.001$. Data represent mean \pm SD ($n = 5$). (d) Detection of IL-1 β , TNF- α , OCN and ALP expression levels in arthritic joints in different groups. Arthritic joints in different groups were stained with IL-1 β , TNF- α and OCN antibodies, respectively. ALP was stained light dark in arthritic joints from different groups (Scale bar = 100 μ m) ($n = 5$).

Supplemental Fig. 11 Quantitative analysis for the immunofluorescence of TUNEL staining in arthritic joints of AIA rats with late-stage arthritis after receiving the indicated treatment. *** $P < 0.001$. Data represent mean \pm SD ($n = 5$).

11) The data in Fig 5D should be quantitated, the statements are not convincingly supported by the pictures.

Response: Per the Reviewer's suggestion, we have quantitated and statistically analyzed the cartilage content of all treated groups. The related results were presented in **Supplemental Fig. 16**.

Supplemental Fig. 16 Quantitative analysis for the safranin O-positive area of ankle joints of AIA rats with advanced arthritis after receiving the indicated treatment. ** $P < 0.01$, *** $P < 0.001$. Data represent mean \pm SD ($n = 5$).

12) What is the mechanism for cleavage of the MMP9 peptide in the joint? Is MMP9 expressed? MMP9 was added to the in vitro activated macrophages. How do we know that the MMP9 target peptide is cleaved in the joint?

Response: Previous studies illustrated that MMP9 was highly expressed in arthritic joints of RA. (*Arthritis Rheum.*, 1996, 39:1576-1587; *J. Clin. Immunol.*, 2006, 26:299-307) Our MMP9 staining results also revealed the increased MMP9 expression in arthritic joints from AIA rats. (**Fig. 1a**) Therefore, the MMP9 target peptide could be cleaved by MMP9 in arthritic joints.

13) How much cell death or other changes occurs in the off target organs of rats treated with CEL-PRNP?

Response: Per the Reviewer's comment, we have conducted TUNEL immunostaining assays to investigate the apoptosis level of major organs (heart, liver, spleen, lung, kidney and brain) after CEL-PRNPs treatment. As shown in **Supplemental Fig. 17**, CEL-PRNPs induced negligible off-target apoptosis in major organs.

In addition, we have conducted serum enzyme assays and histopathological analyses to investigate the *in vivo* safety of CEL-PRNPs. As shown in **Supplemental Fig. 18**, CEL significantly increased serum levels of ALT, AST, CK and LDH in rats, suggesting the toxicity of CEL to liver and heart. CEL-RNPs also remarkably increased serum levels of ALT and AST, demonstrating the liver toxicity of CEL-RNPs to liver. H&E staining results revealed that CEL caused obvious pyknosis of neuron in the brain, atrophy of myocardial cells and myofibrillar loss in the heart, loss of hepatic cords and dilatation of blood sinus in the liver. CEL-RNPs caused severe atrophy, loss of hepatic cords and dilatation of blood sinus in liver tissue. CEL-PRNPs caused slight damages in liver tissue. However, CEL-PRNPs caused no significant changes in serum enzyme levels and displayed no obvious damages to heart, spleen, lung, kidney and brain. Therefore, CEL-PRNPs had good *in vivo* safety and could reduce the neurotoxicity, cardiotoxicity and hepatotoxicity of CEL.

Supplemental Fig. 17 TUNEL immunofluorescence staining of major organs (heart, liver, spleen, lung, kidney and brain) were processed 2 days after rats receiving the indicated treatment. (Scale bar = 50 μm) ($n = 3$).

Supplemental Fig. 18 Safety evaluation. (a-d) The serum level of ALT (a), AST (b), CK (c) and LDH (d) in rats receiving the indicated treatment. $*P < 0.05$, $**P < 0.01$. Data represent mean \pm SD ($n = 5$). (e) Hematoxylin and eosin (H&E) staining of the major organs (heart, liver, spleen, lung, kidney and brain) were processed 2 days after rats receiving the indicated treatment. (magnification 200 \times) ($n = 5$).

14) A control using PRNP, without CEL, should be included in the study. What is the in vivo effect of the nanoparticles?

Response: Per the Reviewer's suggestion, the related experiment has been carried out. Blank PRNPs showed negligible effects on reducing swelling in ankle joints and paws, which was comparable to saline group.

Therapeutic efficacy of blank PRNPs in AIA rats with advanced arthritis. Paw thickness (a) and ankle diameter (b) of AIA rats were recorded every other day during the treatment period. Data represent mean \pm SD ($n = 7$).

15) The sentence beginning on line 151 is confusing and should be rewritten.

Response: Per the Reviewer's suggestion, we have checked and revised our manuscript.

16) Figure 2C is not mentioned in the text, although it probably would go on line 157.

Response: We appreciate the Reviewer's comment and have added the related data in the revised manuscript.

17) Why were murine macrophages used for in vitro studies, rather than rat?

Response: According to previous studies, murine macrophages have been widely used to gain OCs. (*Nature*, 2000, 408:600-605; *Nat. Med.*, 2004, 10:617-624) Thus, we followed the protocol of gaining OCs based on the methods described in their papers.

Reviewer #2:

This manuscript aims for delivering celastrol (CEL) to selectively induce apoptosis of both types of cells (osteoclasts and macrophages) in RA joints. While this is a novel work and nothing wrong with the science, it lacks the required impact to be published as it is in Nat Comm Journal. The work is mostly focused on in vivo studies using AIA rat model, but would be most impactful if validation is done using human cells obtained from patients with RA (e.g. ex vivo models). The fate of PLGA NPs should also be clarified by providing bioelimination and cell internalization (dynamic of endocytosis/exocytosis) data. In addition, some refinement is required with respect to the manuscript contents. Grammar revision should be once performed. Include scale bars in all figures. Some references are also incomplete. In brief, this manuscript requires major revision prior its acceptance for publication.

Response: We appreciate the Reviewer's comments and suggestions. We have collected peripheral blood samples and synovial tissues from patients with late-stage RA undergoing joint replacement surgery. Activated macrophages were isolated from the synovial tissues and osteoclasts (OCs) were generated through stimulating peripheral blood mononuclear cells (PBMCs) with MCS-F and RANKL. We have studied the distribution behavior and apoptosis-inducing ability of CEL-PRNPs on both types of cells. Coumarin 6 (C6) was used as a fluorescent probe and was loaded into prepared nanoparticles. As shown in **Fig. 4**, confocal images and flow cytometry results showed C6-RNPs and C6-PRNPs (in presence of MMP9) had higher drug distribution in both types of cells. In addition, CEL-RNPs and CEL-PRNPs (in presence of MMP9) triggered the higher apoptosis rate of OCs and activated macrophages derived from patients with late-stage RA, as compared to CEL-NPs and CEL-PRNPs in the absence of MMP9. In summary, PRNPs increased drug distribution and cellular apoptosis on both OCs and activated macrophages derived from RA patients by RGD mediated endocytosis after responding to MMP9. These observations were added and presented in our revised manuscript.

Previous studies demonstrated RGD had high affinity to integrin and RGD mediated endocytosis significantly increased the distribution of drug loaded nanoparticles in integrin-positive cells. (*Nat. Biotechnol.*, 1997, 15:542-546. *J. Control. Release*, 2015, 102(1):191-201.) In this study, taking advantage of the RGD-integrin interaction, CEL was efficiently delivered into OCs and activated macrophages, thus inducing increased cellular apoptosis. Confocal imaging and flowcytometry assay revealed different drug distribution among various prepared nanoparticles on target cells and normal cells. The results have proved that both RNPs and PRNPs (with MMP9) selectively delivered drug to OCs and activated macrophages (**Fig. 3 and Fig. 4**) rather than normal cells including non-activated macrophages (BMMs) and normal vascular endothelial cells (HUVECs), (**Fig. 4a, d and Supplemental Fig. 8**) suggesting a good targeting ability of PRNPs to OCs and activated macrophages.

To investigate the fate of prepared PRNPs after *in vivo* intravenous injection into AIA rats, we have studied the biodistribution behaviors of PRNPs in major organs and inflamed joints at different time points. As shown in **Supplemental Fig. 10**, the PRNPs distribution in major organs (heart, liver, lung and kidney) had significantly decreased within 48 h after injection, demonstrating the clearance of prepared PLGA nanoparticles in the body. We also have studied the *in vivo* safety of CEL-PRNPs. Results revealed that CEL-PRNPs had negligible off-target toxicity and significantly reduced the neurotoxicity, cardiotoxicity and hepatotoxicity of CEL. (**Supplemental Fig. 17 and Supplemental Fig. 18**) Given the cell-targeting ability and good safety of PRNPs, the dynamic endocytosis/exocytosis of nanoparticles was not investigated in this study.

Per the Reviewer's suggestion, we have checked the scale bars and magnification of presented figures. In addition, we have proofread our manuscript and corrected the errors in the revised manuscript accordingly.

Fig. 4 Increased apoptosis of OCs and activated macrophages derived from patients diagnosed with late-stage RA by PRNPs *in vitro*. (a) Confocal images (magnification 400 ×) of the cellular uptake on OCs and activated macrophages. (b-d) Quantitative analysis of the cellular uptake of C6-loaded NPs, RNPs or PRNPs on OCs (b), activated macrophages (c) and HUVECs (d) after 1-h incubation at the C6 concentration of 50 ng/mL. ** $P < 0.01$, *** $P < 0.001$. Data represent mean \pm SD ($n = 3$). (e, f) Quantitative analysis for the apoptosis of OCs (e) and activated macrophages (f) by CEL-RNPs, CEL-RNPs or CEL-PRNPs. ** $P < 0.01$, *** $P < 0.001$. Data represent mean \pm SD ($n = 3$).

Supplemental Fig. 10 (a) *Ex vivo* DiD fluorescence images in major organs of AIA rats with advanced arthritis at 2 h, 8 h, 24 h and 48 h post-injection with free DiD, or labeled NPs, RNPs or PRNPs ($n = 3$). (b-f) The statistical graphs of the fluorescence intensity of organs based on the semi-quantitative analysis of the *ex vivo* fluorescence images after *i.v.* administration of free DiD or DiD-labeled nanoparticles. Data represent mean \pm SD, * $P < 0.05$, ** $P < 0.01$, *** $P < 0.001$.

Supplemental Fig. 18 Safety evaluation. (a-d) The serum level of ALT (a), AST (b), CK (c) and

LDH (d) in rats receiving the indicated treatment. * $P < 0.05$, ** $P < 0.01$. Data represent mean \pm SD ($n = 5$). (e) Hematoxylin and eosin (H&E) staining of the major organs (heart, liver, spleen, lung, kidney and brain) were processed 2 days after rats receiving the indicated treatment. (magnification 200 \times) ($n = 5$).

Reviewer #3 (Remarks to the Author):

This is a manuscript that focuses on a novel method of delivering celestrol for suppressing arthritis. The bulk of the manuscript has been published by other groups looking at AIA model and celestrol treatment of macrophages and osteoclasts. The main novelty is the delivery vehicle. However, to demonstrate the efficacy of the prodrug, the authors should use alpha v Beta 3 intern receptor KO mice and MMP9-/- mice.

Response: We appreciate the Reviewer's positive comment with respect to the novelty of the present study. Also, we agree that it is a rational strategy to use knockout mice as negative controls. In our study, the *in vitro* results (including cellular uptake and cell apoptosis assays on activated macrophages and osteoclasts (OCs) derived from mice and RA patients) have demonstrated that CEL-PRNPs could target activated macrophages and OCs through RGD-integrin interaction after responding to MMP9. **(Fig. 3 and Fig. 4)** Furthermore, the *in vivo* results (including biodistribution, *in vivo* antiarthritic mechanism and pharmacodynamic assays) also have shown that PRNPs could be selectively delivered to activated macrophages and OCs in inflamed joints and trigger apoptosis of these cells, **(Fig. 6 and Fig. 7)** consequently controlling inflammation and reversed bone erosion. **(Fig. 8 and Fig. 9)** The above results have proved the efficacy of our strategy. Thus, currently, we did not carry out relevant experiments in knockout mice. However, *in vivo* studies in knockout mice model will be advantageous to the clinical translation of this drug delivery platform and we intend to apply this technology in our future translational research of CER-PRNPs. This information has been added into the "DISCUSSION" section as a limitation of the current study in the revised manuscript.

REVIEWER COMMENTS

Reviewer #1 (Remarks to the Author):

The authors have worked very hard to provide new information. Despite the fact that there are subsets of macrophages in RA synovial tissue that are activated, there are many that are not. I refer the authors to F. Zang et al, Nat Immun, 2019. I am certain that the same thing occurs in animal models of RA. therefore I do not accept the authors assertions about targeting activated macrophages. Most macrophages will take up nanoparticles, and it is possible that only those that are activated express MMP9 and will therefore be subject to the therapeutic effects. This has not been documented.

Some remaining concerns include 1) Even magnified, I cannot see anything in some panels (maybe my computer)

2) What cells are TUNEL+ in Fig 11.

3) supplemental figure 1 misrepresents the data.

4) only some of the macrophages isolated from RA synovial tissue will be activated.

Reviewer #2 (Remarks to the Author):

The authors have addressed the major concerns that have been raised and thus the manuscript has reached the quality and soundness to be published.

Reviewer #1 (Remarks to the Author):

The authors have worked very hard to provide new information. Despite the fact that there are subsets of macrophages in RA synovial tissue that are activated, there are many that are not. I refer the authors to F. Zang et al, Nat Immun, 2019. I am certain that the same thing occurs in animal models of RA. therefore I do not accept the authors assertions about targeting activated macrophages. Most macrophages will take up nanoparticles, and it is possible that only those that are activated express MMP9 and will therefore be subject to the therapeutic effects. This has not been documented.

Response: We appreciate the reviewer's comment and have changed "targeting activated macrophages" to "targeting synovial macrophages" throughout the manuscript. In addition, we have cited the reference mentioned by the reviewer (*F. Zhang et al, Nat Immun, 2019*).

Some remaining concerns include 1) Even magnified, I cannot see anything in some panels (maybe my computer)

Response: We agree that several figures had weak fluorescent signals due to the lower targeting efficacy of free fluorescent probe compared with fluorescent probe labeled nanoparticles. In our safety evaluation study, several organs showed negligible TUNEL signals in saline treated rats compared with different CEL formulations treated rats. We have carefully checked our revised manuscript and we can make sure that the contents and figures in the current study are appropriately presented.

2) What cells are Tunel+ in Fig 11.

Response: Per the reviewer's comment, we have conducted the immunostaining assay to investigate the apoptosis of OCs and macrophages in synovial tissues from AIA rats treated with different nanoparticles. Macrophages and OCs were determined by immunofluorescence analysis of CD68 and CD51. As shown in **Supplemental Fig 11**, CEL-PRNPs was the most efficient in inducing cellular apoptosis in inflamed joints among all treatment groups. Furthermore, the apoptotic cells induced by CEL-PRNPs were mainly synovial macrophages and OCs.

Supplemental Fig. 11 TUNEL immunofluorescence staining in ankle joints from AIA rats receiving the indicated treatment. Macrophages and OCs were determined by immunofluorescence analysis of CD68 and CD51 (red fluorescence), respectively. (Scale bar = 20 μ m), ($n = 3$).

3) supplemental figure 1 misrepresents the data.

Response: We appreciate the reviewer’s comment and have removed the supplemental figure 1 from our revised manuscript.

4) only some of the macrophages isolated from RA synovial tissue will be activated.

Response: We agree with reviewer that only some of the macrophages isolated from RA synovial tissue will be activated. Such information has been added in the “INTRODUCTION” section of the revision. Accordingly, we have changed “targeting activated macrophages” to “targeting synovial macrophages” throughout the manuscript.

Reviewer #2 (Remarks to the Author):

The authors have addressed the major concerns that have been raised and thus the manuscript has reached the quality and soundness to be published.

Response: We appreciate the reviewer’s positive comment.

REVIEWERS' COMMENTS

Reviewer #4 (Remarks to the Author):

1. There is more and more evidence that synovial macrophages (i.e. the resident yolk-sack derived population) are NOT inflammatory but regulatory (see Culemann Nature 2019, Alivernini Nat Med 2020). These data are largely ignored by the authors and instead they talk about "synovial macrophages" as one population. I guess data can be better explained if considering that the drug preferentially targets bone marrow derived inflammatory macrophages. Comparative avb3 expression between BM-derived macrophages and resident macrophages would have been interesting to see and could explain it.

2. Of note, it can't be ruled out that the effect observed is based on targeting neutrophils as they also express avb3 (e.g. see Rainger 1999). It does not seem that the authors have done much effort to rule out this possibility.

Minor: The authors use the term "RA" (e.g. Figure 1) which is incorrect as they do not investigate the human disease, but a model of inflammatory arthritis.

Minor: Reference selection is sometimes odd. Too many reviews cited, which do not directly refer to the project but on the other hand important literature on macrophages in arthritis is missing.

Reviewer #4 (Remarks to the Author):

1. There is more and more evidence that synovial macrophages (i.e. the resident yolk-sack derived population) are NOT inflammatory but regulatory (see Culemann Nature 2019, Alivernini Nat Med 2020). These data are largely ignored by the authors and instead the talk about "synovial macrophages" as one population. I guess data can be better explained if considering that the drug preferentially targets bone marrow derived inflammatory macrophages. Comparative avb3 expression between BM-derived macrophages and resident macrophages would have been interesting to see and could explain it.

Response: We thank the reviewer for the insightful and thoughtful comments. We agree that CEL-PRNPs preferentially targets bone marrow (BM) derived inflammatory macrophages in RA synovium. Both murine activated macrophages (LPS activated) and human synovial macrophages (CD14 positive) used in this study are BM derived cells and show inflammatory properties. In addition, non-activated macrophages show low expression level of $\alpha v\beta 3$ integrin, while BM-derived inflammatory macrophages express high levels of $\alpha v\beta 3$ integrin. (*Ann. Rheum. Dis.*, 2002, 61:ii96. *Nat. Rev. Rheumatol.*, 2012, 8:719-728. *Am. J. Physiol.-Cell Ph.*, 2010, 299:C1267-C1276. *Proc. Natl. Acad. Sci. USA*, 1996, 93:9764) Accordingly, we have changed “targeting synovial macrophages” to “targeting inflammatory macrophages” throughout the manuscript.

2. Of note, it can't be ruled out that the effect observed is based on targeting neutrophils as they also express avb3 (e.g. see Rainger 1999. It does not seem that the authors have done much effort to rule out this possibility.

Response: We appreciate the reviewer's comment. Neutrophils are abundant in both synovial tissue and fluid in the early stage of RA and are important target in the treatment of early RA. (*Arthritis Res. Ther.*, 2010, 12:R196. *Rheum. Dis. Clin. North. Am.*, 1995, 21:691-714) However, we aimed to develop CEL-PRNPs to treat advanced RA. Macrophages represent one potential key mediator in RA joint inflammation and high numbers of macrophages are a prominent feature of inflammatory lesions in established RA. (*Nat. Rev. Rheumatol.*, 2016, 12:472-485. *Arthritis Res. Ther.*, 2000, 2:189. *Arthritis. Rheum.*, 1996, 39:115-124.) Therefore, in this study, the anti-inflammatory effect of CEL-PRNPs were mainly mediated by inducing targeting apoptosis of macrophages.

Minor: The authors use the term "RA" (e.g. Figure 1) which is incorrect as they do not investigate the human disease, but a model of inflammatory arthritis.

Response: Per the reviewer's suggestion, we have revised the legend of Figure 1 in our revised manuscript.

Minor: Reference selection is sometimes odd. Too many reviews cited, which do not directly refer to the project but on the other hand important literature on macrophages in arthritis is missing.

Response: Per the reviewer's suggestion, we have updated the references in our revised manuscript.